# Systematic genetic analysis of the MHC region reveals mechanistic underpinnings of HLA type associations with disease

Matteo D'Antonio[1,2†], Joaquin Reyna[2,3†], David Jakubosky[3,4], Margaret KR Donovan[4,5], Marc-Jan Bonder[6], Hiroko Matsui[1], Oliver Stegle[6], Naoki Nariai[2‡], Agnieszka D'Antonio-Chronowska[1,2], Kelly A Frazer[1,2*]

[1]Institute for Genomic Medicine, University of California, San Diego, San Diego, United States; [2]Department of Pediatrics, Rady Children's Hospital, University of California, San Diego, San Diego, United States; [3]Biomedical Sciences Graduate Program, University of California, San Diego, La Jolla, United States; [4]Bioinformatics and Systems Biology Graduate Program, University of California, San Diego, San Diego, United States; [5]Department of Biomedical Informatics, University of California, San Diego, San Diego, United States; [6]European Molecular Biology Laboratory, European Bioinformatics Institute, Cambridge, United Kingdom

**Abstract** The MHC region is highly associated with autoimmune and infectious diseases. Here we conduct an in-depth interrogation of associations between genetic variation, gene expression and disease. We create a comprehensive map of regulatory variation in the MHC region using WGS from 419 individuals to call eight-digit HLA types and RNA-seq data from matched iPSCs. Building on this regulatory map, we explored GWAS signals for 4083 traits, detecting colocalization for 180 disease loci with eQTLs. We show that eQTL analyses taking HLA type haplotypes into account have substantially greater power compared with only using single variants. We examined the association between the 8.1 ancestral haplotype and delayed colonization in Cystic Fibrosis, postulating that downregulation of *RNF5* expression is the likely causal mechanism. Our study provides insights into the genetic architecture of the MHC region and pinpoints disease associations that are due to differential expression of HLA genes and non-HLA genes.

**\*For correspondence:**
kafrazer@ucsd.edu

[†]These authors contributed equally to this work

**Present address:** [‡]Illumina, Inc, San Diego, United States

**Competing interests:** The authors declare that no competing interests exist.

## Introduction

The 4 Mb major histocompatibility (MHC) region on chromosome 6p21.3, which encodes the human leucocyte antigen (HLA) gene complex, is highly polymorphic, gene dense, and has strong linkage disequilibrium (LD) (*Dendrou et al., 2018*; *Fehrmann et al., 2011*). Genome-wide association studies (GWAS) have identified many autoimmune and infectious diseases that are more prevalent in individuals carrying certain HLA types (alleles), whereas other HLA types are protective against disease (*Blackwell et al., 2009*; *Gough and Simmonds, 2007*; *Matzaraki et al., 2017*). The molecular mechanisms by which certain HLA types predispose or protect against disease are currently unclear. Several studies have suggested that the mechanistic underpinnings underlying associations with some autoimmune diseases may be due to certain HLA types being more likely than other HLA types to atypically present self-antigens (*Klein et al., 2014*; *Oldstone, 1998*; *Yin et al., 2013*). Expression differences between different HLA types of the same HLA gene could also contribute to disease associations. Finally, some of the genetic associations with specific HLA types could be due to causal variants on the same haplotype, that affect the expression or function of non-HLA genes in the MHC region (*Holoshitz, 2013*). Hence there are complex interactions between genetic variation, gene

expression and function of both HLA and non-HLA genes, and disease, rendering the MHC region an important but difficult interval to genetically interrogate.

Here, we used deep whole genome sequence (WGS) from 419 individuals to call full resolution (eight-digit) HLA types and RNA-seq data from 361 matched induced pluripotent stem cells (iPSCs) to create a comprehensive map of regulatory variants in the MHC region. We identified eight-digit HLA types that are strongly associated with expression differences of their cognate HLA gene, and conducted eQTL analyses using both single-variants and eight-digit HLA types, which represent haplotypes of coding and regulatory variants. We built upon the comprehensive MHC regulatory map by functionally exploring GWAS signals for 4083 complex traits, finding strong evidence of colocalization for 180 disease loci with eQTLs. Our data are a valuable resource for both the genetics and disease-focus communities, providing insights into the genetic architecture of the MHC region that have not previously been described. To demonstrate the utility of our resource, we examined the common Caucasian 8.1 ancestral haplotype (8.1AH), which spans the entire MHC region and is associated with many autoimmune disorders and other diseases (*Gambino et al., 2018*), but is protective against bacterial colonization in Cystic Fibrosis (CF) patients (*Tomati et al., 2015*). Here, we show that 8.1AH is associated with decreased expression of *RNF5*, and previous studies have shown that inhibition of RNF5 results in rescue of F508del mutant cystic fibrosis transmembrane conductance regulator (CFTR) function (*Sondo et al., 2018*). Based on these observations, we postulate that downregulation of *RNF5* expression is the causal mechanism underlying the association between 8.1AH and delayed colonization in CF patients.

## Results

### Eight-digit HLA typing

We used HLA-VBSeq (*Nariai et al., 2015*) to estimate HLA types at eight-digit resolution for 30 genes [six HLA class I genes (HLA-A, -B, C, -E, -F, -G); eight HLA Class I pseudogenes (HLA-H, -J, -K, -L, -P, -T, -V, -W); twelve HLA class II genes (HLA-DMA, -DMB, -DOA, -DOB, -DPA1, -DPA2, -DPB1, -DPB2, -DQA1, -DQB1, -DRA, -DRB1); and four non-HLA genes (MICA, MICB, TAP1, and TAP2)] in 650 WGS samples (from 419 unique individuals) and identified 526 unique alleles (*Figure 1*, *Supplementary file 1*, *Supplementary file 2*). The 650 WGS samples were obtained by combining data from two large induced pluripotent stem cell (iPSC) resources: 1) iPSCORE (273 individuals) (*Panopoulos et al., 2017*), and 2) HipSci (377 samples from 146 individuals) (*Kilpinen et al., 2017a*; *Kilpinen et al., 2017b*; *Streeter et al., 2017*). All the WGS samples had deep coverage and the expected high polymorphism rate across the MHC region (*Figure 2A–B*). The 146 individuals in HipSci were of European descent while the 273 iPSCORE individuals are of diverse ethnic backgrounds (190 European, 30 Asian, 20 Multiple ethnicities, 18 Hispanic, 7 African American, 5 Indian and 3 Middle Eastern) and thus harbored HLA alleles from multiple human populations (*Supplementary file 3*). While all 146 HipSci donors were unrelated, there were 183 donors in iPSCORE that are part of 56 unique families (2–14 individuals/family), including 25 monozygotic (MZ) twin pairs. For 84 HipSci individuals, we HLA typed 231 matched fibroblast-iPSC WGS pairs (one fibroblast sample and two or more iPSC clones per donor).

### HLA types have high recall rates and reproducibility

For each HLA gene, we initially calculated recall rate, measured as the fraction of individuals that could be HLA-typed. We observed a high recall rate for all 30 genes (mean = 98.5% for both iPSCORE and HipSci samples, *Figure 2C,D*) and for only three HLA genes in iPSCORE (*HLA-H, HLA-T* and *HLA-K*) and four genes in HipSci (*HLA-H, HLA-T, HLA-K* and *HLA-DRB1*) fewer than 95% of individuals were HLA-typed. The fraction of heterozygous and homozygous HLA types across the 30 genes was highly similar in the iPSCORE and HipSci individuals (r = 0.92). These results show that we were able to obtain high recall rates for all 30 MHC region genes.

To examine the reproducibility of the eight-digit HLA types, we determined HLA type concordance across the 25 MZ twin pairs (defined as twin concordance), Mendelian inheritance concordance from 17 non-overlapping trios and HLA type concordance of the 231 fibroblast-iPSC pairs (defined as HipSci concordance). Overall, we found that the median concordance across for all 526 HLA alleles was very high; however, both the twin pairs (96.7%) and fibroblast-iPSC pairs (95.7%)

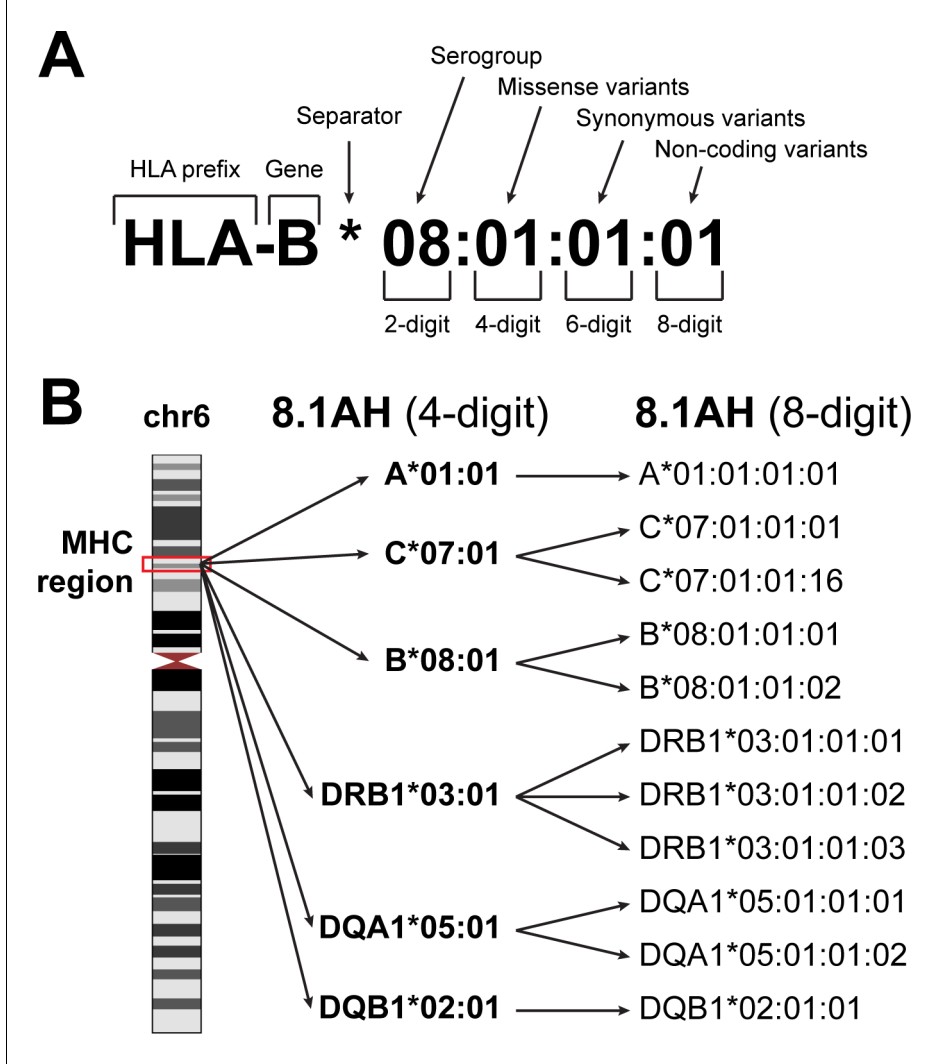

**Figure 1.** HLA nomenclature. (**A**) Description of an HLA type (allele) using current standard nomenclature. The first two digits typically correspond to the serological antigen. The third and fourth digits distinguish HLA types that differ by one or more missense variants. The fifth and sixth digits distinguish HLA types that differ by one or more synonymous variants. The seven and eight digits distinguish HLA types that differ by noncoding variants. (**B**) The location of the MHC region on chromosome 6 (left). The 8.1 ancestral haplotype (8.1AH) is comprised of six HLA types (corresponding to six HLA genes) typically reported at four-digit resolution (middle). For each of the six HLA genes, we identified one to three eight-digit resolution HLA types that can be collapsed into their 8.1AH four-digit counterparts (right).

had slightly higher concordance than the Mendelian inheritance (90.9%, *Figure 2E,F,G*). These results show that HLA-VBSeq produces highly reproducible results when samples with exactly the same two HLA alleles are analyzed; but has slightly lower reproducibility calling the same HLA alleles when in different diplotypes.

To estimate the accuracy of the 526 called HLA alleles we tested for Hardy-Weinberg equilibrium (HWE) in the unrelated individuals and found that the vast majority (511, 97.1%) were in HWE ($p > 1 \times 10^{-6}$, *Figure 2—figure supplement 1A–C*), consistent with the reproducibility values in twin pairs (96.7%) and in fibroblast-iPSC pairs (95.7%). We also conducted a comparative analysis with HLA alleles in 3.5 million individuals with diverse genetic backgrounds from the Allele Frequency Net Database (AFND) (*González-Galarza et al., 2015*) and observed high correlation between HLA allele frequencies in AFND and iPSCORE (r = 0.921) and HipSci (r = 0.908, *Figure 2—figure supplement 1D*). Overall, HLA-VBSeq (*Nariai et al., 2015*) was able to accurately detect eight-digit

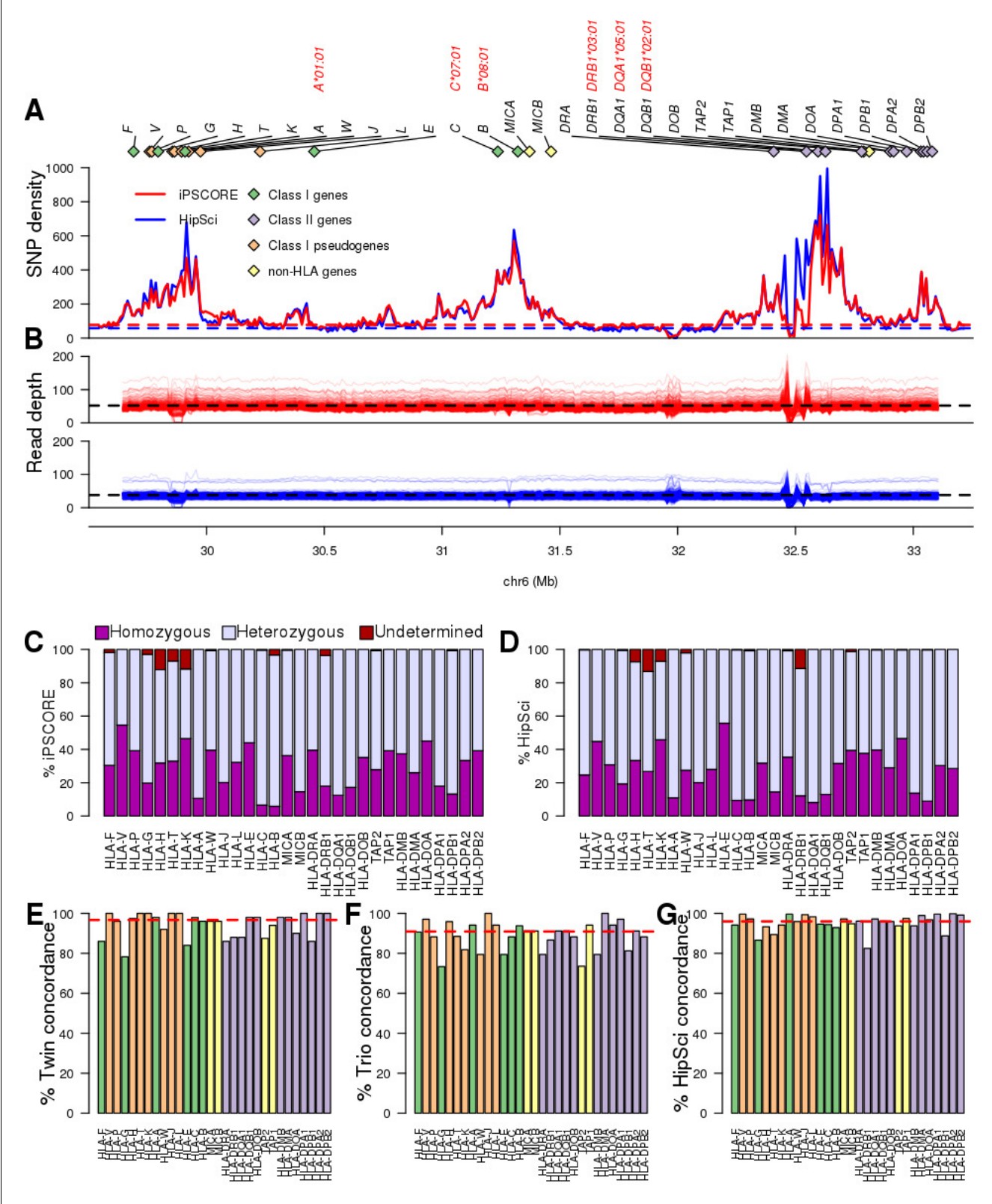

**Figure 2.** WGS coverage in MHC region and HLA types. (**A**) SNP density distribution in consecutive 10 kb bins in the HLA region (chr6: 29690551, 33102442). The dashed lines represent the chromosome-wide SNP density in the iPSCORE (blue) and HipSci (red) WGS samples. The 30 HLA genes analysed in this study are represented by diamonds with green representing HLA Class I, purple HLA Class I pseudogenes, orange HLA Class II and yellow non-HLA genes. The four-digit HLA types for the six genes composing the 8.1AH are shown in red. (**B**) Read depth coverage of the HLA region

*Figure 2 continued on next page*

*Figure 2 continued*

for the iPSCORE WGS samples (top, red) and the HipSci WGS samples (bottom, blue) was calculated using consecutive 10 kb bins. Each line represents a single WGS sample, the black dashed lines represent the average coverage for the iPSCORE WGS samples (52X) and the HipSci WGS samples (38X). (C, D) Fraction of the 273 iPSCORE (C) and 377 HipSci (D) WGS samples with homozygous, heterozygous and undetermined alleles for each of the 30 HLA genes. For each sample, if one of the two alleles for an HLA gene was undetermined, we considered the HLA genotype as 'undetermined'. (E) Twin concordance (iPSCORE: 25 monozygotic twin pairs), (F) Mendelian inheritance concordance (iPSCORE: 17 trios) and (G) Fibroblast-iPSC concordance (HipSci: 231 fibroblast-iPSC pairs) of HLA types at eight-digit resolution. Genes were sorted based on their genomic position and colored according to their class, as shown in panel A. The red dashed line represents the median across all genes.

The online version of this article includes the following figure supplement(s) for figure 2:

**Figure supplement 1.** Allele frequencies of predicted HLA types.

resolution HLA alleles in the 650 WGS samples as evidenced by high recall rates, reproducibility and allele frequencies consistent with Hardy-Weinberg law.

## Eight-digit HLA types associated with expression of cognate HLA gene

To analyze gene expression, we used RNA-seq data from 446 iPSC samples (derived from 361 of the HLA typed individuals *Supplementary file 2*). There were 146 expressed genes in the MHC interval (TPM >2 in at least 10 samples), of which 24 were HLA genes, including six HLA class I genes, 3 HLA class I pseudogenes, 11 HLA class II genes, *MICA*, *MICB*, *TAP1* and *TAP2*. For each individual, we estimated expression levels of the 24 HLA genes by aligning reads to cDNA sequences specific for the HLA types called by HLA-VBSeq (*Nariai et al., 2015*) to avoid alignment biases (*Aguiar et al., 2019*; *Gensterblum-Miller et al., 2018*; *Lee et al., 2018*; *Panousis et al., 2014*) (*Figure 3—figure supplement 1*, *Figure 3—figure supplement 2*, *Supplementary file 4*). The HLA class I genes tended to be expressed at high levels, consistent with their ubiquitous expression in all cell types, and the HLA class II genes tended to be expressed at lower levels, as expected due to their primary role in immune cells (*Matzaraki et al., 2017*). To examine the extent that HLA types at eight-digit resolution were associated with gene expression levels, we compared the expression levels of each HLA type against all other alleles from the cognate HLA gene. We quantile-normalized expression of each gene and compared the TPM distributions between all samples carrying a specific HLA allele and all the other samples (*Figure 3A,B*). We observed high variability in the expression levels of the different HLA alleles, with 189 of 428 HLA types (44.2%) present in at least two individuals showing a significantly different expression level of the cognate HLA gene compared with the samples not carrying the HLA type (t-test, FDR < 5%, *Supplementary file 5*). Five HLA genes (*HLA-A*, *HLA-C*, *HLA-DRB1*, *HLA-DQB1* and *HLA-DPB1*) showed more than four-fold differences between the least expressed and the most expressed alleles. We examined the HLA types corresponding to the 8.1AH at eight-digit resolution (*Figure 1B*), and determined that the class I genes tended to be expressed higher than the other HLA types of the cognate gene, while two of the class II genes (*HLA-DQA1* and *HLA-DQB1*) were significantly expressed lower and one (*HLA-DRB1*) was not differentially expressed (*Figure 3C–H*). Overall, eight-digit HLA types were strongly associated with expression differences of the cognate HLA gene, suggesting that the associations between HLA types and disease may in part be driven by differential allelic expression.

## eQTLs associated with the expression of 56 eGenes in the MHC region

We performed an eQTL analysis by testing each of the 146 expressed genes and the genotypes of 45,245 variants (MAF >1%) spanning the MHC region (±1 Mb flanking each side) using a linear mixed model. We detected associations between 56 eGenes and 72,473 eQTLs (Benjamini-Hochberg-corrected p-value<0.1, *Supplementary file 6*) corresponding to 26,363 distinct variants; hence 36.4% of the variants in the MHC region were associated with the expression of one or more gene. Due to the low recombination rate (*DeGiorgio et al., 2014*; *Trowsdale and Knight, 2013*) and complex LD structure (*Jensen et al., 2017*; *Miretti et al., 2005*) in the MHC region, we investigated the distributions of all primary eQTLs rather than solely focusing on lead variants (*Figure 4*). The 56 eGenes tended to be associated with high numbers of primary eQTLs (median = 842), ranging from 26 (*HLA-E*) to 4911 (*HLA-DQB1*). Many of the eGenes were also associated with primary eQTLs spanning relatively large distances (median = 2.1 Mb), including *ZFP57*, *HLA-C*, *HLA-B*, *MICA* and *RNF5*,

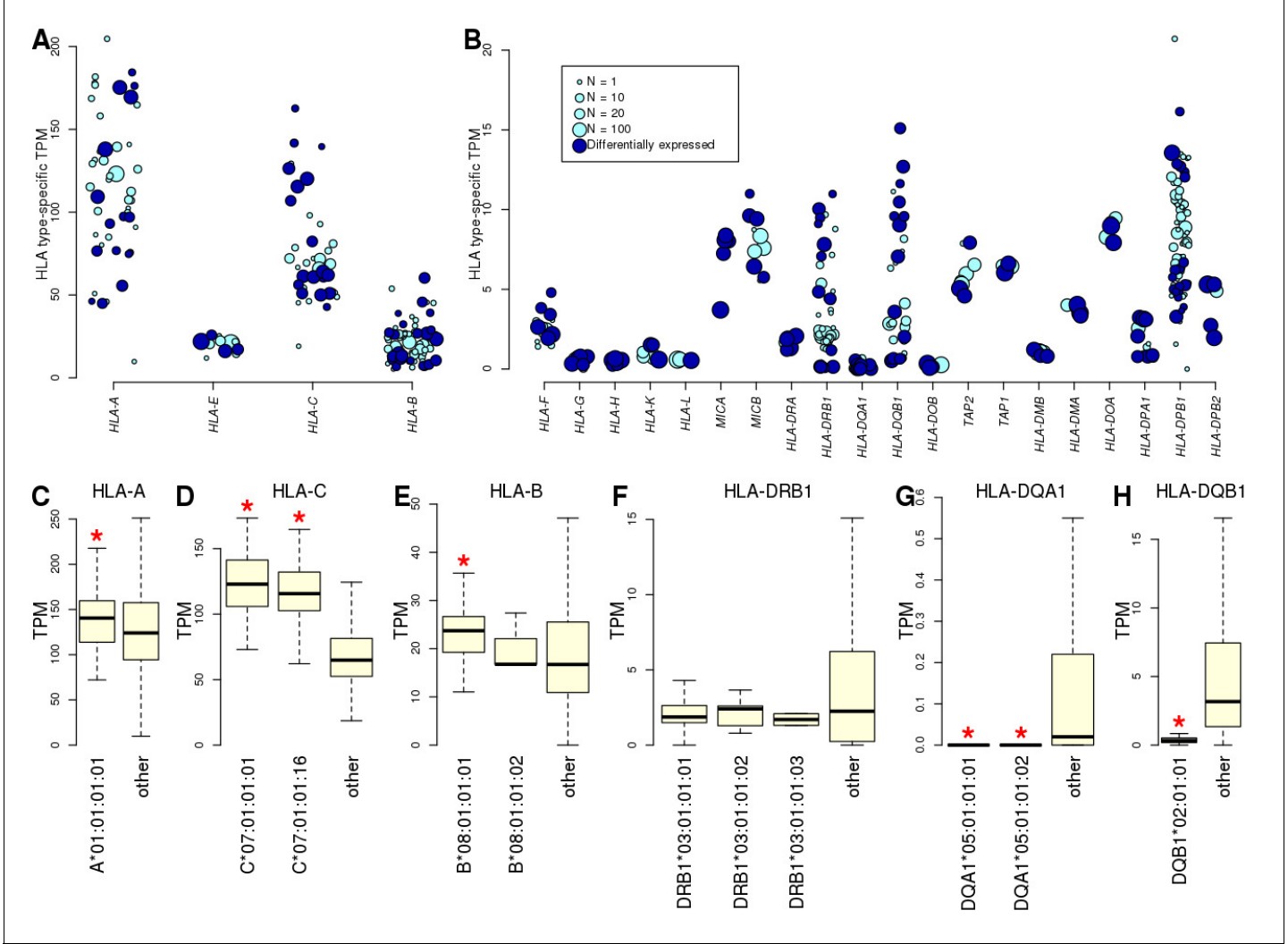

**Figure 3.** Allele-specific expression of HLA genes in iPSCs. (A, B) Mean TPM expression across HLA types: (A) shows the four HLA genes with highest expression levels (mean TPM >20); and (B) shows lower expressed HLA genes (mean TPM <20). Each point represents one allele. Point size represents the number of individuals carrying each allele. All HLA types are shown, although differential gene expression analysis (**Supplementary file 5**) was performed only on those present in at least two individuals. Eight-digit HLA-types significantly associated with the expression of their cognate HLA gene are shown in dark blue. (C–H) Boxplots showing mean expression of the eight-digit 8.1AH HLA types (**Figure 1**) compared with the mean expression of all the other HLA types for the cognate gene. Red stars indicate that the mean expression of the ancestral haplotype HLA type is significantly different (**Supplementary file 5**).

The online version of this article includes the following figure supplement(s) for figure 3:

**Figure supplement 1.** Comparing reference and HLA type-specific TPM.

**Figure supplement 2.** HLA genes – Reference versus HLA type-specific expression.

whose associated primary eQTLs nearly spanned the entire 4 Mb MHC region, which is consistent with the strong LD of ancestral haplotypes. While genome-wide eQTL analyses usually test associations between gene expression and neighboring variants (within 500 kb or 1 Mb), our results show that gene expression in the MHC is associated with *cis* regulatory variants that span regions several times larger than those normally tested.

To identify independent eQTLs, we repeated the analysis conditioning gene expression on the lead eQTL for each of the 56 eGenes. For 42 eGenes, we observed secondary eQTLs (conditional on the lead eQTL), and for 36 eGenes we identified a tertiary eQTL upon conditioning on the top two independent eQTLs (**Figure 4**, **Figure 5**). Overall, for the 56 eGenes, we observed 76,809 eQTLs (72,473 primary, 2424 conditional 1 and 1912 conditional 2, Bonferroni-adjusted p-value<0.1) with

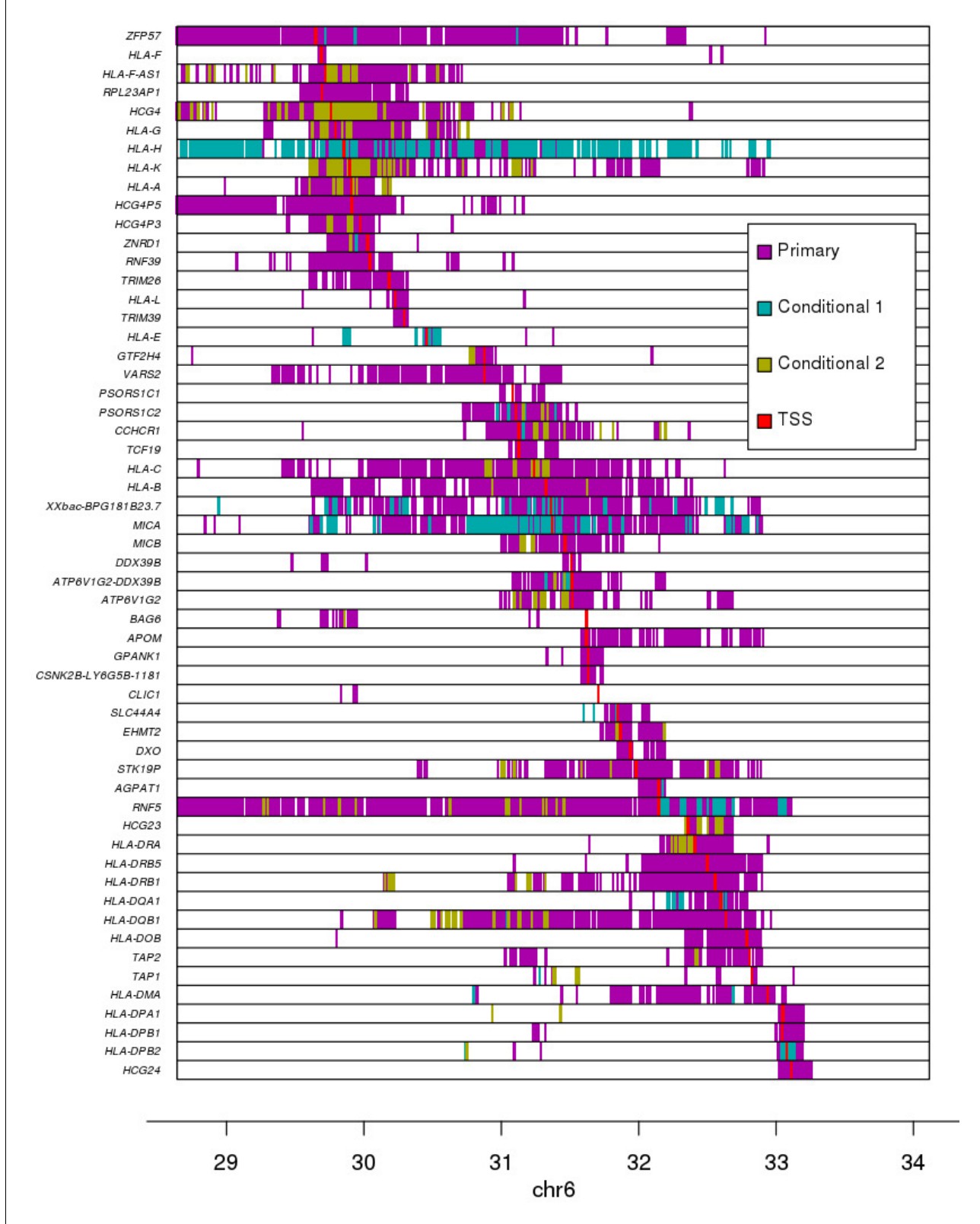

**Figure 4.** Associations between variants and gene expression levels in the MHC region. eQTL associations for 56 genes in the MHC region. Colors represent whether each eQTL is primary or conditional: 'conditional 1' refers to eQTLs independent from the lead eQTL and 'conditional 2' refers to eQTLs independent from the lead eQTL and the lead 'conditional 1' eQTL. TSS for each gene is shown in red.
*Figure 4 continued on next page*

*Figure 4 continued*

The online version of this article includes the following figure supplement(s) for figure 4:

**Figure supplement 1.** Distance of eQTLs from TSS.
**Figure supplement 2.** Lead eQTL associations for each eGene.
**Figure supplement 3.** Enrichment of eQTLs in regulatory elements.
**Figure supplement 4.** Comparing ASE between eGenes and non-eGenes.
**Figure supplement 5.** eGenes with heterozygous eQTL variants are enriched for ASE.

genome-wide significance corresponding to 27,200 distinct variants. These results show that, within the gene-dense MHC region, a large fraction (56; 38.4%) of the expressed genes are eGenes, many

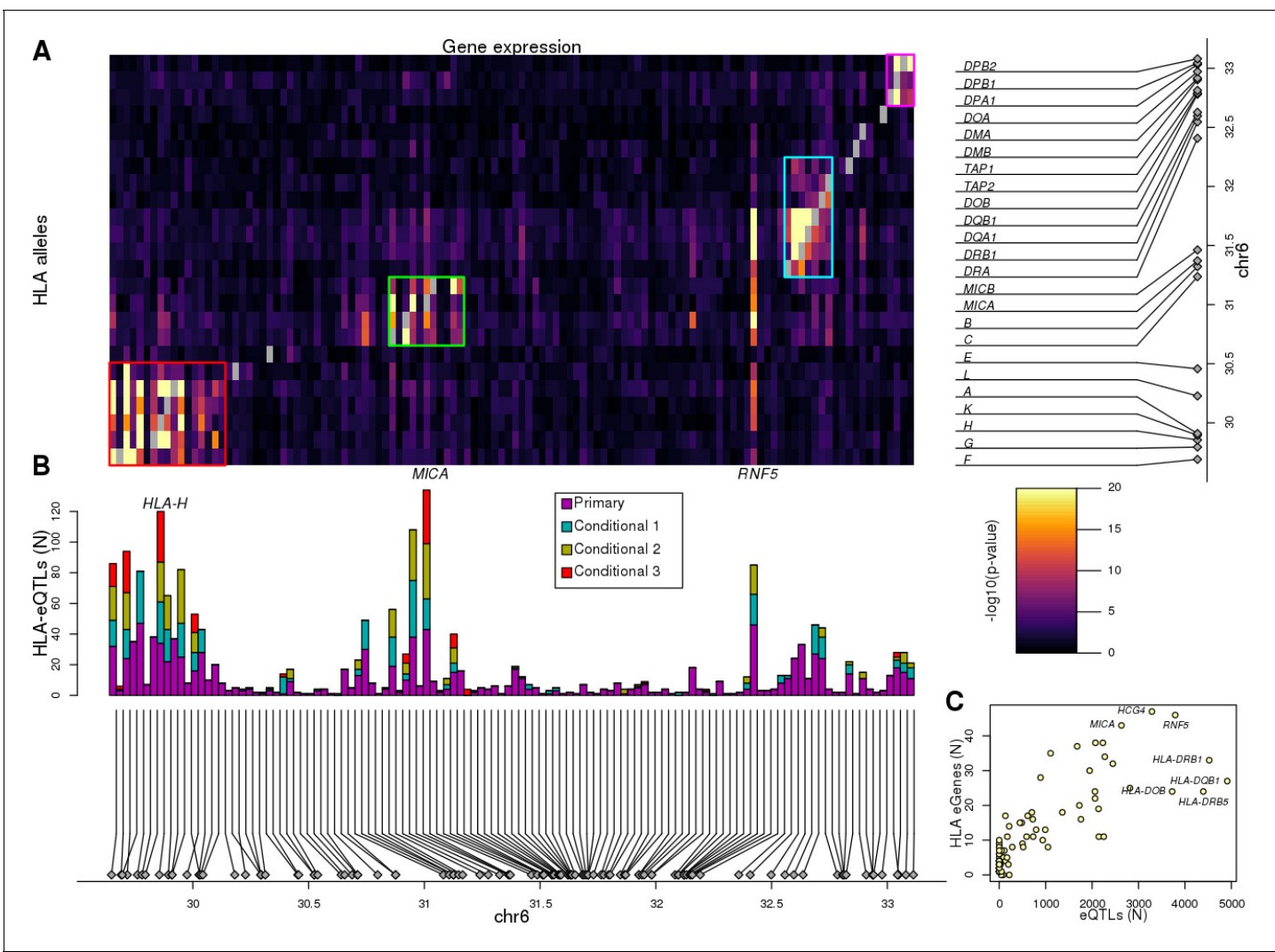

**Figure 5.** Associations between HLA types and gene expression levels in the MHC region. (**A**) Heatmap showing the associations between HLA types at 8-digit resolution (Y axis) and the expression of 114 genes in the MHC region (X axis). For eGenes associated with multiple HLA-type eQTLs, only the most significant p-value is shown. Self-associations between the expression of each HLA gene and its HLA types were not considered (grey squares). Four different groups of genes with shared HLA-eQTLs are outlined in different colors. (**B**) Barplot showing for each of the 114 genes in the heatmap in (**A**), the number of primary and conditional associations (adjusted based on the lead eQTL, the top two independent eQTLs, or the top three independent eQTLs) between HLA types and gene expression levels. 'Conditional 1' refers to HLA-eQTLs conditional on the lead primary SNP eQTL, 'Conditional 2' refers to HLA-eQTLs conditional on the lead primary and lead conditional 1 eQTLs and 'Conditional 3' refers to HLA-eQTLs conditional on the lead primary, conditional 1 and conditional 2 eQTLs. (**C**) Scatterplot showing the number of primary eQTLs (X axis) and the number of primary HLA-type eQTLs (Y axis) for the 52 genes that had both eQTLs and HLA-type eQTLs.

of which (42; 75%) are associated with multiple independent regulatory variants that span long distances.

Given the complexity of conducting genetic association studies in the MHC region, we set out to validate the eQTLs that we discovered by showing that they have characteristics of regulatory variants. We initially investigated the distance between lead eQTLs and the transcription start site (TSS) of their associated eGene and found that in general they were close (median distance = 18.3 kb, *Figure 4—figure supplement 1*), and for 13 eGenes (23.2%), the lead eQTL was localized within 5 kb of their TSS. We next visually examined the association between the expression levels of each of the 56 eGenes and genotypes of the lead eQTL and observed a strong correlation (*Figure 4—figure supplement 2*). We examined if the 76,809 primary and conditional eQTLs (27,200 distinct variants) were enriched in regulatory elements by testing their overlap with chromatin states in iPSC lines from the Roadmap Epigenomics Consortium (*Kundaje et al., 2015*). We observed that the eQTLs in the MHC region with the largest effect sizes were enriched for overlapping regulatory elements active in iPSCs including active transcription start sites (TSS), regions flanking active TSS, UTRs (defined as regions transcribed at 5′ and 3′) and transcribed regions (*Figure 4—figure supplement 3*). To further characterize the associations between eQTLs and gene expression, we tested if the 56 eGenes were more likely to have allele-specific expression (ASE) than the other 90 expressed MHC genes; and found that eGenes were significantly more likely to have ASE [mean allelic imbalance fraction (AIF) = 67.3%, compared with 61.9% in the other genes, p=4.38×10$^{-64}$, Mann-Whitney U test, *Figure 4—figure supplement 4*, *Supplementary file 7*]. We also observed that eGenes in samples with heterozygous eQTLs (primary and conditional) have stronger ASE compared with eGenes in samples homozygous for eQTLs (39,891 eQTLs in total examined, corresponding to 21,739 variants; *Figure 4—figure supplement 5*). These results show that the primary and conditional eQTLs in the HLA region have characteristics of regulatory variants including proximity to the TSS of their associated eGene, enrichment in active chromatin marks, and association with allele-specific expression of their eGene.

## Regulatory variants in the MHC region play important roles in complex traits

To characterize the associations between genetic variation in the MHC region, gene expression and disease, we investigated if eGenes and complex traits shared the same causal variants using a colocalization approach (*Giambartolomei et al., 2014*). We tested the eQTL signals (primary and conditional) for all 56 MHC eGenes against the GWAS signals for 4083 complex traits from the UK Biobank and as expected identified many strong associations (1480 loci with posterior probability of association (PPA) >0.2 for sharing the same causal variant, including 180 with PPA >0.8, *Supplementary file 8*).

To demonstrate the utility of using our resource, we explored colocalization between eGenes and complex traits that contribute to wide phenotypic variation in Cystic Fibrosis (CF) patients including pulmonary diseases (*Stoltz et al., 2015*), weight (*Leung et al., 2017*; *Morison et al., 1997*) as well as liver, biliary tract and gallbladder diseases (*Assis and Debray, 2017*; *Diwakar et al., 2001*; *Herrmann et al., 2010*) (*Figure 6*). We identified eight eGenes whose eQTL colocalized with GWAS signals for pulmonary diseases, of which three, *HLA-B*, *HLA-DQA1* and *HLA-DQB1*, had previously associated with Asthma in GWAS (*Kontakioti et al., 2014*; *Li et al., 2012*; *Mahdi et al., 2018*) and in the UK Biobank (*Vicente et al., 2017*). However, the other five eGenes, including *HLA-DRA* (the most strongly associated), *APOM*, *DXO*, *RNF5* and *TAP1* were novel, that is had not been identified in GWAS. Interestingly, the association of *RNF5* was consistent with its function as a regulator of the CFTR protein and suggested role in cystic fibrosis (*Sondo et al., 2018*; *Tomati et al., 2015*). We next identified four novel eGenes (*EHMT2*, *GPANK1*, *HLA-C* and *RNF5*) associated with weight phenotypes. The two eGenes most strongly associated, *GPANK1* and *RNF5*, had opposite associations. *GPANK1* was only associated with fat-free mass, suggesting that the protein product is involved in muscle or bone mass regulation. Conversely, *RNF5* was more strongly associated with fat mass than fat-free mass, suggesting that its altered expression can contribute to CF phenotypic variation, as these individuals are known to have bile and other digestive issues, including difficulties in assimilating fat (*Bijvelds et al., 2005*; *Freudenberg et al., 2008*; *Morison et al., 1997*; *Wilke et al., 2011*). Finally, eQTLs for *HLA-DQB1*, *PSORS1C2* and *RNF5* colocalized with liver and gallbladder GWAS signals. These findings suggest that altered gene expression levels in the MHC region are

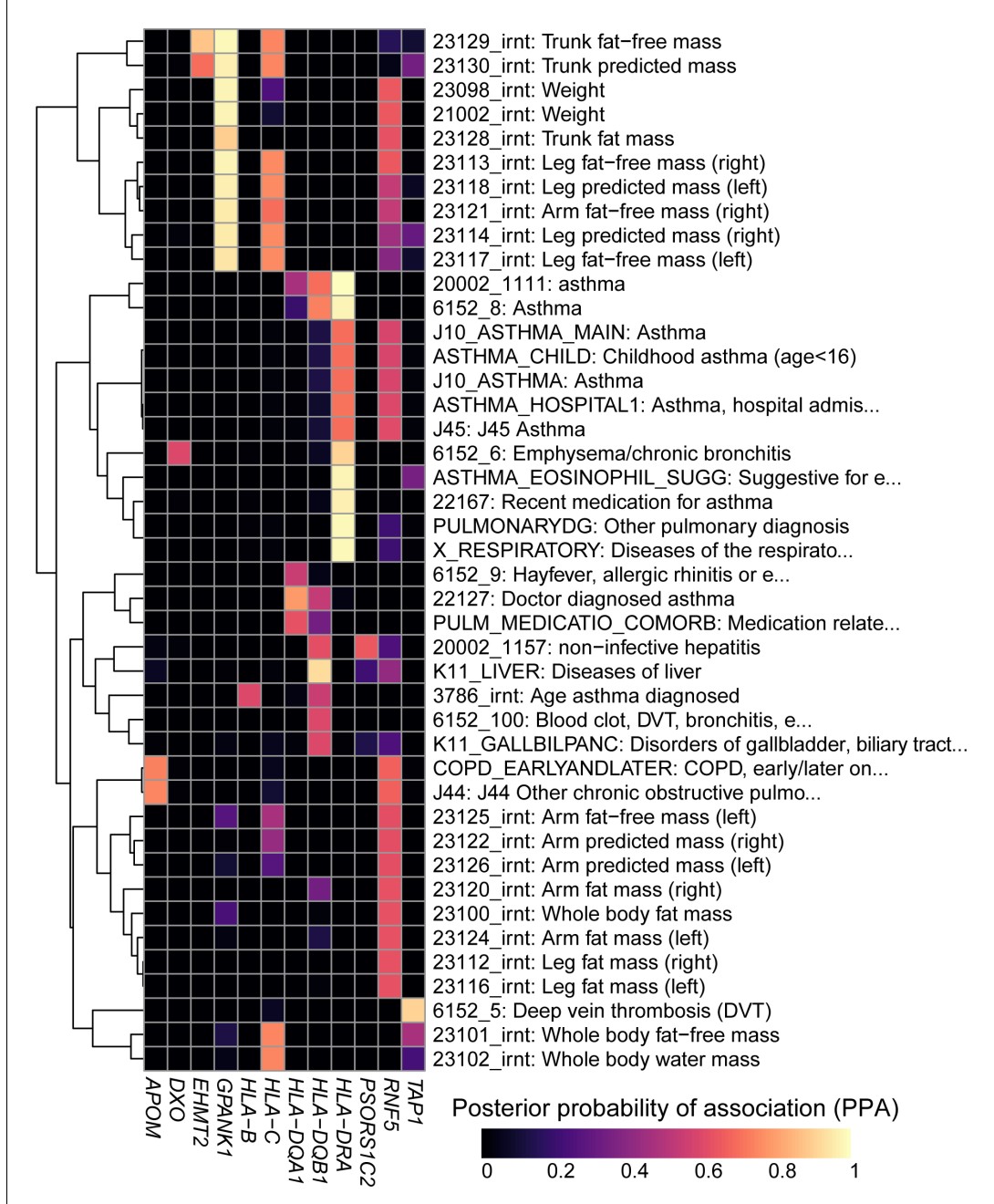

**Figure 6.** Colocalization results. Heatmap showing the posterior probability of association (PPA) for colocalization (*Giambartolomei et al., 2014*) between an eQTL for the indicated eGene (columns) and a GWAS signal for the indicated trait (rows). GWAS traits in *Supplementary file 8* were filtered to include those having at least one eGene with PPA > 0.5 and the following terms in their description: 'asthma', 'pulmonary', 'mass', 'weight', 'fat', 'liver diseases' and/or 'biliary tract diseases'. Associations with PPA > 0.2 for all 4,083 UKBB traits are reported in *Supplementary file 8*.

associated with phenotypic variation in CF patients. They also demonstrate the utility of our data to serve as a resource to narrow down the location of causal regulatory variants underlying known associations between genes in the MHC region and complex traits, as well as to identify biologically meaningful novel associations.

## HLA-type eQTLs associated with the expression of 114 eGenes in the MHC region

Given that eight-digit HLA types represent a haplotype of many regulatory and coding SNPs (*Figure 1*), we examined if conducting an eQTL analysis using HLA types would detect regulatory signals independent from those detected in the single-variant eQTL analysis. We calculated associations between each of the 283 common eight-digit HLA-types (MAF $\geq$1% in 361 individuals with expression data) and the 146 genes expressed in the HLA region, a linear mixed model (LMM) (see Materials and methods). We first examined self-associations between the expression of each HLA gene and its HLA types (for example, we tested the associations between *HLA-A* expression and HLA-A:01:01:01:01). We found that for all 24 expressed HLA genes, there were at least two alleles significantly associated (referred to as HLA-type eQTLs hereafter, Benjamini-Hochberg-corrected p-value<0.05) with its expression (median = 4, range 2 to 11; *Supplementary file 9*), which is consistent to what we observed in the allele-specific expression analysis (*Figure 3*). Self-associations were not considered for the remaining HLA-type eQTL analyses. We detected 1139 associations between 250 HLA-type eQTLs and 114 eGenes (*Supplementary file 9*, *Figure 5A,B*). Associations primarily occurred between HLA-type eQTLs and the expression of genes located in their proximity (median distance = 146.2 kb for significant associations compared with 1.35 Mb for non-significant associations, p=$2.7\times10^{-67}$, Mann-Whitney U test), resulting in HLA-type eQTLs largely clustering into four distinct genomic regions. Interestingly, *RNF5* showed strong associations with multiple HLA types in clusters 1, 2 and 3, confirming the observation that it had long-range associations with eQTLs spanning the entire MHC region (*Figure 4*). In total, there were 62 eGenes identified only by the HLA-type eQTL analysis, four identified only by the single-variant eQTL analysis and 52 identified by both analyses (44.4% all 118 eGenes; *Supplementary file 10*). These results show that the HLA-type eQTL analysis had substantially greater power to identify eGenes than the single-variant eQTL analysis.

We sought to determine if HLA-type eQTLs provided independent signals from the single-variant eQTL analysis in the 52 overlapping eGenes. We observed that the number of HLA-type eQTLs was highly correlated with the number of eQTLs (r = 0.829, p=$7.9\times10^{-31}$, *Figure 5C*). To identify HLA-type eQTL independent signals, we conditioned gene expression on the lead eQTLs from the single-variant analysis. We observed 406 HLA-type eQTLs conditional on the lead eQTLs (49 eGenes), 316 HLA-type eQTLs conditional on the top two independent eQTLs (27 eGenes) and 150 HLA-type eQTLs (13 eGenes) conditional on the top three independent eQTLs (*Figure 5B*, *Supplementary file 9*). Of note, the conditional HLA-eQTLs were largely confined to the four genomic regions containing clusters of genes with shared HLA-eQTLs and the *RNF5* gene. These findings show that the HLA-eQTL analysis identifies a different set of regulatory variants than the single-variant eQTL analysis even for the 52 eGenes discovered by both analyses.

## HLA-type eQTLs capture regulatory variation associated with expression of neighboring genes

We investigated the molecular underpinnings of the clustering of the most significant HLA-type eQTLs within the four distinct genomic regions (*Figure 5A*): 1) chr6:29640168–30152231, including HLA class I genes (*HLA-A*, *HLA-F* and *HLA-G*) and HLA class I pseudogenes; 2) chr6:31171745–31512238, including HLA class I genes (*HLA-B*, *HLA-C*), *MICA* and *MICB*; 3) chr6: 32361463–32789609, including HLA class II genes (*HLA-DRA*, *HLA-DRB1*, *HLA-DQA1*, *HLA-DQB1* and *HLA-DOB*), *TAP1* and *TAP2*; and 4) chr6:32916389–33115544, including HLA class II genes (*HLA-DPA1*, *HLA-DPB1*, and *HLA-DPB2*). We observed that different eGenes in each cluster were typically associated with the genotypes of different HLA types, suggesting that the observed clustering was not due to co-expression (i.e., correlated expression levels) of neighboring HLA genes (*Figure 7*). There was one exception in cluster 3, with the expression of both *HLA-DRB5* and *HLA-DRB1* associated with many of the same HLA types (*Figure 7C*). For this reason, we examined the co-expression of the eGenes in each of the four clusters. Only five eGenes in cluster 3 (*HLA-DRA*, *HLA-DRB5*, *HLA-DQA1*, *HLA-DQB1* and *HLA-DRB1*) displayed moderate correlated expression (*Figure 7—figure supplement 1*), despite having their expression levels largely associated with different HLA types of neighboring genes (*Figure 7*). These findings suggest that the HLA-type eQTLs capture regulatory variants associated with the expression of one or a few neighboring genes. Furthermore, unlike the

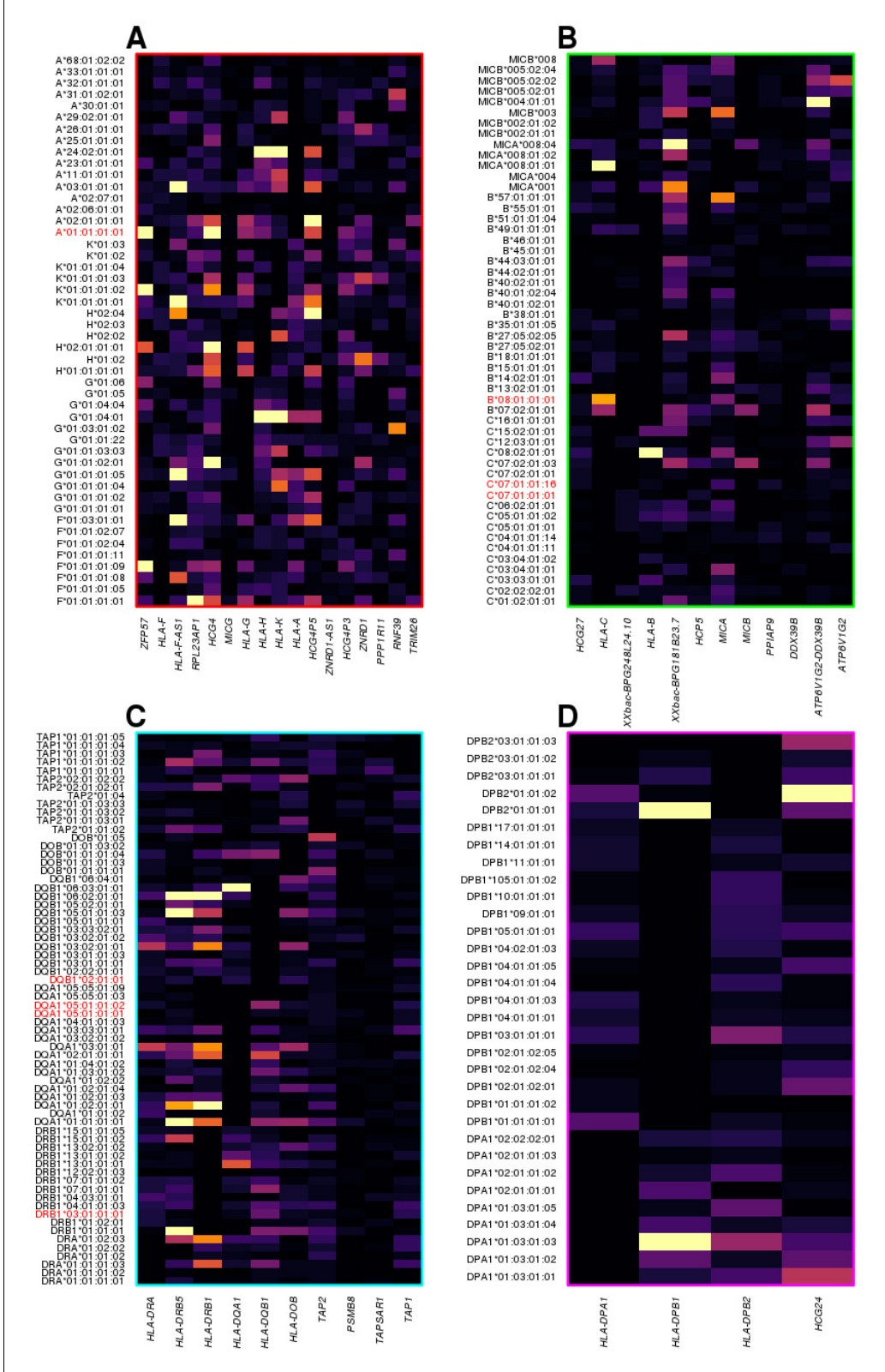

**Figure 7.** HLA-type eQTLs for HLA genes in the four clusters shown in *Figure 6*. Heatmaps showing, for each of the four clusters identified in *Figure 6*, the associations between HLA-type eQTLs and gene expression. Each column represents an eGene, while rows represent all HLA-type eQTLs associated with other genes in the cluster. 8.1AH HLA types are highlighted in red. The heatmaps show that the HLA-type eQTLs capture regulatory variants associated with the expression levels of neighboring genes.

*Figure 7 continued on next page*

*Figure 7 continued*

The online version of this article includes the following figure supplement(s) for figure 7:

**Figure supplement 1.** Coexpression analysis of the MHC region.

eQTLs from the single-variant analysis that tend to span long distances most likely due to the strong LD in the MHC region (*Figure 4*), the HLA-type eQTLs capture regulatory variants that act over relatively short distances.

## *RNF5* expression levels underlie association between 8.1AH and colonization in CF

While we were able to colocalize the single-variant eQTLs and GWAS signals (*Figure 6*), this was not possible for HLA-type eQTLs, because GWAS for complex traits have not yet been conducted using eight-digit HLA types. Therefore, we decided to perform focused analyses of associations between HLA-type eQTLs, *RNF5*, and the 8.1AH to gain insight into the molecular mechanisms underlying CF phenotypic variation. The expression of *RNF5* was negatively associated with eight-digit HLA types for all six genes composing the 8.1AH (*Figure 8A–F*, *Figure 8—figure supplement 1*, *Supplementary file 6*). While we discovered several eight-digit resolution HLA types for some of the HLA genes composing 8.1AH (*Figure 1*), only one eight-digit HLA type for each of the six genes was associated with *RNF5* expression. It has been previously proposed that the association between 8.1AH and delayed bacterial colonization in CF patients could be due to the impact of the ancestral HLA-types on the microbiota composition in the lungs (*Laki et al., 2006*). However, our findings, combined with the fact that RNF5 is a current drug target in CF patients because its downregulation contributes to stabilizing and rescuing the function of mutant CFTR proteins (*Sondo et al., 2018*; *Tomati et al., 2015*), suggests that the association between 8.1AH CF carriers and delayed bacterial colonization could be due to decreased expression of *RNF5* (*Lyczak et al., 2002*; *Mall and Hartl, 2014*; *McNicholas, 2017*; *Pier et al., 1996*). In this proposed model, CF patients carrying 8.1AH express *RNF5* at lower levels than non-carriers, resulting in lower protein levels of RNF5 and thereby less degradation of the misfolded mutated CFTR protein, improved Cl⁻ secretion, lower mucus secretion and delayed colonization by *S. aureus* and *P. aeruginosa* (*Figure 8F*).

## Discussion

The vast majority of risk loci identified through GWAS are located in non-coding regions and the causal variants are implicitly assumed to affect gene regulation. Colocalization methods integrating GWAS and eQTL studies can identify causal variants that are shared between a complex trait locus and altered gene expression; and thereby elucidate the mechanism underlying GWAS non-coding variants and identify their target genes. Although the human MHC region on 6p21.3 has been associated with many autoimmune and infectious diseases, it is typically excluded from eQTL studies due to its high SNP frequency and complex LD structure (*Lam et al., 2017*). To address this gap, we developed a comprehensive map of regulatory variation in the MHC region using deep WGS from 419 individuals and RNA-seq data from matched iPSCs. We have previously shown that iPSCs are well-powered for eQTL mapping (*Bonder et al., 2019*; *Jakubosky et al., 2019a*; *Jakubosky et al., 2019b*), have a distinct regulatory landscape relative to somatic tissues (*DeBoever et al., 2017*), and thereby provide insights into regulatory variants that exert their effects during early development. While we demonstrate the feasibility of generating a map of regulatory variation for the MHC region and its utility for examining molecular mechanisms underlying disease associations in the interval, future studies using eQTLs from other cell types could increase the number of complex trait loci in the interval that are functionally annotated.

We constructed a large panel of HLA types at eight-digit resolution for 30 genes in the MHC region using a state-of-the-art computational algorithm, HLA-VBSeq (*Nariai et al., 2015*), and the comprehensive IPD-IMGT/HLA database (*Robinson et al., 2016*). To examine the extent that eight-digit HLA types were associated with expression levels of their cognate HLA gene, we estimated expression levels of the HLA genes by aligning RNA-seq reads to cDNA sequences specific for the HLA types called by HLA-VBSeq. Overall, we observed that the eight-digit HLA types were strongly

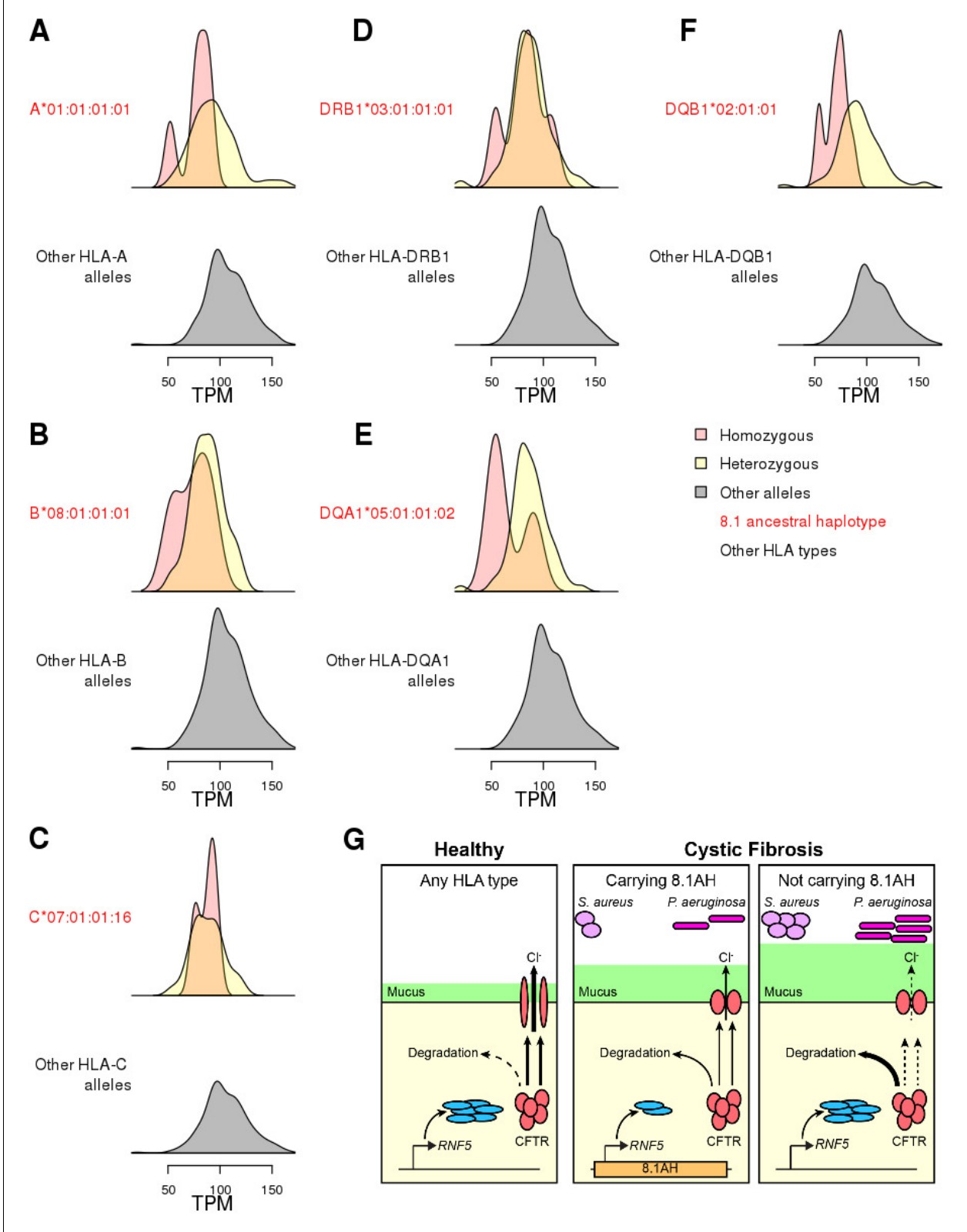

**Figure 8.** Associations between HLA-type eQTLs, *RNF5* expression and Cystic Fibrosis. (**A–F**) Distributions of *RNF5* expression levels in samples that are homozygous (red) and heterozygous (yellow) for significant HLA-type eQTLs corresponding to 8.1AH HLA alleles. *RNF5* expression distributions for all samples that do not carry the indicated 8.1AH HLA alleles are shown in grey. *RNF5* expression distributions for all significant HLA-type eQTLs that are not part of the 8.1AH are shown in *Figure 8—figure supplement 1*. (**G**) Cartoon depicting proposed model of the molecular mechanisms

*Figure 8 continued on next page*

*Figure 8 continued*

underlying the associations between 8.1AH, *RNF5* expression and CFTR function in CF. On the left, in healthy individuals, CFTR is correctly folded in the airway epithelium and Cl⁻ ions can be secreted. On the right, in CF patients not carrying 8.1AH, CFTR is misfolded and high levels of RNF5 result in its degradation which, in turn, results in decreased Cl⁻ secretion, mucus hypersecretion and colonization by *S. aureus* and *P. aeruginosa*. In the middle, in CF patients carrying 8.1AH, *RNF5* is expressed at low levels, resulting in the lower levels of RNF5, less degradation of the misfolded mutated CFTR protein, improved Cl⁻ secretion, lower mucus secretion and delayed colonization by *S. aureus* and *P. aeruginosa*.

The online version of this article includes the following figure supplement(s) for figure 8:

**Figure supplement 1.** Association of RNF5 expression levels with HLA-type eQTLs.

associated with expression differences of the cognate HLA gene. This finding suggests that the associations between certain HLA types and disease may not only be driven by polymorphic amino acid epitopes resulting in differential binding affinities to specific peptides, but also in part by different expression levels.

We identified 56 eGenes in the MHC region that were associated with high numbers of primary eQTLs (median = 842) spanning relatively large distances (median = 2.1 Mb), which is consistent with the strong LD in the interval. We also conducted conditional eQTL analyses and discovered that the majority of these eGenes were associated with multiple independent regulatory variants. Given the complexity of conducting genetic association studies in the MHC region, we examined the primary and conditional eQTLs and showed that they have characteristics of regulatory variants including proximity to the TSS of their associated eGene, enrichment in active chromatin marks, and association with allele-specific expression of their eGene. We then tested the eQTL signals (primary and conditional) for all 56 eGenes against GWAS signals for 4083 complex traits from the UK Biobank and found strong evidence of colocalization with 180 disease loci. Given that the 8.1 ancestral haplotype (8.1AH) is an important genetic modifier of lung disease in Cystic Fibrosis (CF) patients (*Laki et al., 2006*), we examined if any of the MHC eGenes colocalize with loci for complex traits contributing to phenotypic variation in CF, including pulmonary diseases (*Stoltz et al., 2015*), weight (*Leung et al., 2017*; *Morison et al., 1997*), as well as liver, biliary tract and gallbladder diseases (*Assis and Debray, 2017*; *Diwakar et al., 2001*; *Herrmann et al., 2010*). Our findings suggest that altered gene expression levels in the MHC region contributes to phenotypic variation in these complex traits and demonstrates the utility of our data to identify shared causal variants underlying GWAS and eQTL signals in the interval.

We hypothesized that HLA types, which represent a haplotype of many regulatory and coding SNPs, could exert a greater effect on gene expression than single variants. We conducted an HLA-type eQTL analysis and identified 114 eGenes, showing that indeed taking HLA type haplotypes into account substantially increased power to identify eGenes over single-variant eQTL analysis. HLA-type eQTLs largely clustered into four distinct genomic regions. We showed that this clustering was due to the fact that HLA-type eQTLs capture regulatory variants that act over relatively short distances and tend to be associated with the expression of only one or a few neighboring genes. Unfortunately, we were not able to conduct a colocalization analysis of HLA-type eQTLs and GWAS signals, because GWAS for complex traits have not yet been conducted using eight-digit HLA types. However, in the past few years, novel methods have been developed to impute HLA types from variants assayed by SNP arrays (*Jia et al., 2013*; *Zheng et al., 2014*), which makes conducting GWAS using HLA types feasible for currently existing datasets. Based on our findings, future GWAS directly testing associations between HLA types and complex traits are likely to uncover many novel associations.

We examined the association between the 8.1AH and delayed colonization in cystic fibrosis (CF) (*Laki et al., 2006*). We showed that *RNF5* expression was negatively associated with eight-digit HLA types for all six genes composing the 8.1AH. Interestingly, RNF5 was recently described as a potential novel drug target in CF, because, when RNF5 activity is decreased, the mutant CFTR protein becomes stabilized and its function as a cAMP-dependent chloride channel in the lung epithelium is partially rescued (*Sondo et al., 2018*; *Tomati et al., 2015*). Impaired ion transport mediated by CFTR is associated with reduced airway hydration and decreased mucociliary clearance, which leads to increased vulnerability to bacterial infection (*Dechecchi et al., 2018*; *Munder and Tümmler, 2015*). Hence, we postulate that downregulation of *RNF5* expression due to the presence of specific

regulatory variants in 8.1AH is the likely mechanism underlying delayed colonization by *S. aureus* and *P. aeruginosa*. While our findings do not preclude that HLA types comprising 8.1AH have a direct role in the protection against bacterial colonization in CF, they suggest that reduced *RNF5* expression likely plays a major factor in the association between 8.1AH and colonization.

Taken together, our data provide insights into the genetic architecture of the MHC region and show that disease associations with HLA types cannot only be due to epitope differences but differential expression of HLA genes and associated non-HLA genes. Further, these data are a valuable resource for both the genetics and disease-focus communities that can be exploited to identify causal mechanisms underlying disease associations in the MHC region, and potentially novel drug targets.

# Materials and methods

## Key resources table

| Reagent type (species) or resource | Designation | Source or reference | Identifiers | Additional information |
|---|---|---|---|---|
| Software, algorithm | GATK | (*Van der Auwera et al., 2013*; *Zhang et al., 2017*) | | |
| Software, algorithm | HLA-VBSeq | (*Nariai et al., 2015*) | | |
| Software, algorithm | SAMtools version 1.2 | (*Li et al., 2009*) | | |
| Software, algorithm | STAR (2.5.0a) | (*Dobin et al., 2013*; *Harrow et al., 2012*) | | |
| Software, algorithm | RSEM (1.2.20) | (*Li and Dewey, 2011*) | | |
| Software, algorithm | bcftools | (*Li et al., 2009*) | | |
| Software, algorithm | lmekin (coxme package) | R 3.5.1 | | |
| Software, algorithm | MBASED | (*Mayba et al., 2014*) | | |
| Software, algorithm | coloc.abf (coloc package) | R 3.5.1 (*Giambartolomei et al., 2014*) | | |

### Whole-genome sequencing data

*iPSCORE Resource*: Whole genome sequencing (WGS) was performed for 273 individuals (152 females and 121 males) at Human Longevity, Inc (HLI) as described in *DeBoever et al. (2017)*. Briefly, DNA was isolated from blood or skin fibroblasts (254 and 19 samples, respectively) using DNEasy Blood and Tissue Kit. A Covaris LE220 instrument was used to quantify, normalize and shear the DNA samples, which were then normalized to 1 μg, DNA libraries were made using the Illumina TruSeq Nano DNA HT kit and their size and concentration determined using LabChip DX Touch (Perkin Elmer) and Quant-iT (Life Technologies), respectively. We normalized all concentrations to 2–3.5 nM, prepared combined pools of six samples and sequenced them at 150 bp paired-end on Illumina HiSeqX. WGS data were generated at an average of 1.1 billion paired-end reads (depth of coverage: 52X) per sample (range 720 million to 3.1 billion). WGS reads were aligned to the reference genome (GRCh37/hg19) including decoy sequences (hs37d5) (*Auton et al., 2015*) using BWA-MEM version 0.7.12 (*Li and Durbin, 2010*). Fastq file quality was estimated using FASTQC (http://www.bioinformatics.babraham.ac.uk/projects/fastqc/). All BAM files were processed using GATK best practices (*Van der Auwera et al., 2013*; *Zhang et al., 2017*), as described in *DeBoever et al. (2017)* to detect single nucleotide polymorphisms (SNPs) genome-wide. For chromosome 6, the chromosome-wide SNP density (79 SNPs per 10 kb) was calculated by dividing the total number of SNPs in the 273 iPSCORE individuals by the length of the chromosome, excluding undefined 'N' nucleotides.

*HipSci Resource*: WGS was performed for 146 individuals (85 females and 61 males) and 231 associated iPSC lines. Fibroblasts obtained from each HipSci individual were reprogrammed and between one to two iPSC lines were derived. We downloaded CRAM files for the 377 samples (146 fibroblast samples and 231 iPSCs) from the European Nucleotide Archive (ENA, https://www.ebi.ac.uk/ena, study PRJEB15299) and transformed to BAM files using Samtools (*Li et al., 2009*). The WGS

data were an average of 835 million paired-end reads (depth of coverage: 38X) per sample (range 450 million to 1.8 billion). BAM files were processed using GATK to detect SNPs on chromosome 6.

## WGS coverage and distribution of variants in MHC region

We assessed the alignment and variant calling of the WGS in the MHC region (chr6:29640168–33115544) by determining SNP density in consecutive 10 kb windows across the region. The SNP density in the WGS data varied widely: 1) for the 273 iPSCORE genomes from 1 SNP/10 kb to 900 SNPs/10 kb (mean = 149.2 SNPs/10 kb) compared with the average of 79.6 SNPs/10 kb across chromosome 6 (*Figure 1A*); and 2) for the 377 HipSci genomes (146 fibroblast samples and 231 iPSCs) from 1 SNP/10 kb to 995 SNPs/10 kb (mean = 155.4 SNPs/10 kb) compared with the average of 58.1 SNPs/10 kb across chromosome 6. The region with the highest SNP density contained the *HLA-DQA1* and *HLA-DQB1* genes, which is consistent with previous studies (*Norman et al., 2017*). To determine whether the high SNP density interfered with read alignment, we analyzed the read depth across the region. We observed that the read depth (mean $49.8 \pm 12.1$ for iPSCORE and $37.9 \pm 7.1$ for HipSci), overall, was comparable with the genome average (52X and 38X, respectively, *Figure 1B*). We found, on average, 98.1% and 96.5% of the MHC region had a read depth >20X, and for each iPSCORE and HipSci sample only $284 \pm 122$ bp and $252 \pm 115$ bp respectively were not covered by any reads.

## HLA typing from whole-genome sequencing data with HLA-VBSeq

HLA-VBSeq (*Nariai et al., 2015*) estimates the most probable HLA types from WGS data at eight-digit resolution by simultaneously optimizing read alignments to a database of HLA type sequences and the abundance of reads on the HLA types by variational Bayesian inference. HLA typing was carried out as follows. First, we identified 30 target genes that had more than one allele at eight-digit resolution in the IMGT/HLA database (release 3.30.0) (*Robinson et al., 2016*), which included six HLA class I genes (HLA-A, -B, C, -E, -F, -G), eight HLA Class I pseudogenes (HLA-H, -J, -K, -L, -P, -T, -V, -W), twelve HLA class II genes (HLA-DMA, -DMB, -DOA, -DOB, -DPA1, -DPA2, -DPB1, -DBP2, -DQA1, -DQB1, -DRA, -DRB1) and four non-HLA genes (MICA, MICB, TAP1, TAP2). Second, for each of the 650 samples (273 iPSCORE and 377 HipSci), reads which aligned to the target genes were extracted from each BAM file using coordinates from Gencode v19 (https://www.gencodegenes.org/releases/19.html) and SAMtools version 1.2 (*Li et al., 2009*). Third, for each sample the extracted reads were re-aligned to the collection of all genomic HLA sequences in the IPD-IMGT/HLA database (release 3.30.0) (*Robinson et al., 2016*), allowing each read to be aligned to multiple reference sequences using the '-a' option in BWA-MEM. The expected read counts for each HLA type were obtained with HLA-VBSeq software (http://nagasakilab.csml.org/hla/). For each HLA gene, only the alleles with mean coverage $\geq$20% of the average coverage calculated over the whole genome were considered, and a target HLA genotype was determined as follows: 1) If no allele passed the threshold, then there were not enough reads aligned to correctly identify an HLA type, and hence no alleles were called; 2) If there was only one allele that passed the threshold, and the depth of coverage was two or more times greater than that of the threshold, then the HLA locus was considered to be homozygous for that HLA allele; 3) If there was only one allele that passed the threshold, and the depth of coverage was less than twice that of the threshold, then the allele was called heterozygous with the second allele not determined; 4) If there were two or more alleles that passed the threshold, the alleles were sorted based on the depth of coverage (from high to low), if the depth of coverage of the top allele was more than twice as that of the second one, then the HLA locus was called homozygous for the top allele; 5) If there were two or more alleles that passed the threshold, the alleles were sorted based on the depth of coverage (from high to low), if the depth of coverage of the top allele was less than twice that of the second one, then the two alleles with the highest coverage were selected as the HLA type. For HLA types at four-digit resolution (used in *Figure 2—figure supplement 1D*) the eight-digit resolution was converted by removing high digit values.

## Determining recall, twin concordance, Mendelian inheritance concordance and fibroblast-iPSC concordance of HLA type detection

Reproducibility of HLA typing for each gene at eight-digit resolution was assessed by calculating: 1) recall, the fraction of determined HLA types for each gene; 2) HLA type concordance in 25 monozygotic twin pairs in the iPSCORE resource; 3) Mendelian inheritance concordance across 17 trios in the iPSCORE resource; and 4) HLA type concordance in 231 fibroblast-iPSC pairs. The recall was calculated independently for the 273 iPSCORE samples and the 377 HipSci samples as the number of determined HLA types divided by the total number of samples. For each gene, we calculated concordance as the fraction of HLA types that matched in the 25 monozygotic twin pairs in iPSCORE or the 231 fibroblast-iPSC paired genomes from the same individual in the HipSci resource. We calculated Mendelian inheritance concordance for each gene as the percentage of HLA types segregating in non-overlapping trios (i.e. in families with multiple trios, each individual was only used once for this analysis). If one or more HLA types were undetermined for a given pair/trio, we excluded the pair/trio during the concordance calculation.

## Hardy-Weinberg equilibrium (HWE)

To investigate HWE, we performed a likelihood ratio test on the allele frequency of each HLA allele independently in 275 unrelated Caucasians in the iPSCORE and HipSci cohorts. We tested each HLA allele independently for several reasons: 1) calculating HWE on sites with tens or hundreds of alleles is highly computationally intensive and complicated (*Eguchi and Matsuura, 1990*; *Graffelman et al., 2017*; *Guo and Thompson, 1992*; *Li et al., 2014*); and 2) the MHC region is under high selective pressure, therefore the proportions between different alleles do not satisfy the assumptions for HWE (*Hardy, 1908*). All associations with p-values$<1\times10^{-6}$ were considered as significantly deviating from HWE and flagged in the HLA-type QTL analysis.

## Comparison of observed HLA type frequencies with other populations

In the Allele Frequency Net Database (AFND) (*González-Galarza et al., 2015*), 3,556,301 people were genotyped for 12 out of the 30 genes we examined in the MHC region at two-digit resolution, 3,469,268 people (17 genes) at four-digit, 124,721 people (14 genes) at six-digit, and 10,212 people (seven genes) at eight-digit. This would result in testing 226 alleles at two-digit resolution (115 iPSCORE and 101 HipSci), 395 alleles at four-digit resolution (222 iPSCORE and 173 HipSci), 321 alleles at six-digit resolution (174 iPSCORE and 147 HipSci), and 42 alleles at eight-digit resolution (21 iPSCORE and 21 HipSci). We conducted the allele frequency comparative analysis using four-digit resolution (identified by collapsing eight-digit HLA types) to maximize the number of individuals in AFND, the number of genes and the number of alleles. The allele frequency of each HLA type in iPSCORE, HipSci and in the AFND was calculated by dividing the total number of individuals containing the given HLA type by the total number of people in each cohort and a correlation value was calculated after fitting a linear model.

## Human iPSCs used in expression studies

For the expression studies, we used RNA-seq data for 446 iPSC samples from 361 individuals (215 iPSCORE and 146 HipSci, *Supplementary file 2*). For the 215 iPSCORE individuals, fibroblasts were reprogrammed, an iPSC clone obtained and RNA-seq data generated and processed as previously described (*D'Antonio-Chronowska et al., 2019*; *D'Antonio et al., 2018*; *DeBoever et al., 2017*; *Panopoulos et al., 2017*). Briefly, sequenced RNA-seq reads were aligned to the Gencode V.19 transcriptome using STAR (2.5.0a) (*Dobin et al., 2013*; *Harrow et al., 2012*). Gene expression levels (TPM) were calculated using RSEM (1.2.20) (*Li and Dewey, 2011*). For the 231 iPSC samples in HipSci (corresponding to 146 individuals), RNA-seq data were downloaded from the European Nucleotide Archive (ENA, https://www.ebi.ac.uk/ena, study PRJEB15299) and processed using the same pipeline used for the iPSCORE iPSC RNA-seq data.

## Estimating HLA type-specific gene expression levels

The IMGT-IPD HLA database release 3.30.0 (*Robinson et al., 2016*), RSEM version 1.2.20 (*Li and Dewey, 2011*) contained cDNA sequences corresponding to each eight-digit resolution allele of the 24 HLA genes expressed in iPSCs. To minimize the number of variant differences between the RNA-

seq reads mapping to the HLA genes, for each individual, we estimated HLA type-specific expression by replacing the canonical cDNA reference sequences of the 24 genes in Gencode v19 with cDNA sequences corresponding to the HLA types detected in the individual using HLA-VBSeq (*Nariai et al., 2015*). Because individuals with heterozygous HLA types had two cDNA sequences associated with the same HLA gene, while homozygous individuals had only one, we determined HLA type-specific expression in two distinct ways. For heterozygous individuals, the TPM associated with each cDNA sequence was considered as HLA type-specific expression, while for homozygous individuals, the TPM was divided by two to obtain HLA type-specific expression levels.

## HLA types associated with expression levels of cognate HLA gene

We compared the expression level distributions of each HLA type against all other alleles from the cognate HLA gene (*Figure 2*; *Supplementary file 5*). Nominal p-values were calculated using Mann-Whitney U test. To FDR-correct, we used the p.adjust function in R with method = 'BH' (Benjamini-Hochberg). All tests with adjusted p-value<0.05 were considered significant.

## eQTL detection

For eQTL analyses, we used the WGS from 361 individuals (215 iPSCORE and 146 HipSci, *Supplementary file 2*) for which iPSC RNA-seq data were available. We filtered variants that: 1) had a call rate <95%; 2) had minor allele frequencies < 1%; 3) deviated from HWE in the 275 unrelated Caucasian individuals ($p<1.0\times10^{-6}$). We obtained 45,245 filtered variants (40,668 SNPs and 4577 small indels) in the MHC region (plus 1 Mb upstream and downstream; exact coordinates: chr6:29640168–33115544) using bcftools (*Li et al., 2009*) and decomposed multiallelic genotypes using vt (*Tan et al., 2015*). Out of the 383 Gencode v19 genes in the MHC locus (chr6:29640168–33115544), 146 were expressed genes (TPM ≥2 in at least 10 samples), including 24 HLA genes. eQTL analysis requires expression levels associated with each gene; therefore, for the 24 HLA genes, we transformed the estimated HLA type-specific expression levels to gene-level expression by summing the expression of both HLA types in each individual. Expression levels of all 146 genes were quantile-normalized across all individuals. Normalized gene expression levels were adjusted for sex, age, batch (iPSCORE and HipSci), iPSC passage, read depth, read length, ten PEER factors (*Stegle et al., 2010*) and, to account for ethnicity, ten principal components calculated on the genotypes of 90,081 common SNPs in linkage equilibrium that we previously used to estimate the ethnicity of iPSCORE individuals (*Panopoulos et al., 2017*). To adjust for potential unknown sources of noise in gene expression, the ten PEER factors were calculated on genome-wide expression levels of 20,595 genes (with TPM ≥2 in at least 10 samples), rather than only on the expression levels of genes in the MHC region. We performed eQTL analysis for the 146 expressed genes and the 45,245 variants using a linear mixed model (lmekin from the coxme package in R) with a kinship matrix (to take population structure into account, including multiple ethnicities, members of the same family and twin pairs) calculated on 90,081 common SNPs distributed across the genome and in linkage equilibrium.

In order to detect significant eQTLs for each eGene, and then detect eGenes at genome-wide significance levels, we employed a three-step procedure to perform hierarchical multiple testing correction (*Huang et al., 2018*): 1) locally adjusted p-value at gene-level: nominal p-values were Bonferroni-corrected for the number of tests performed for each gene (45,245, *Supplementary file 6A*); the variant with the minimum adjusted p-value was considered as lead eQTL (*Supplementary file 6B*); 2) genome-wide adjusted p-value of lead eQTLs: to FDR-correct across all expressed genes, we further adjusted the p-values of the lead eQTLs using the p.adjust function in R with method = 'BH' (Benjamini-Hochberg) and n = 20,595 (corresponding to the number of expressed genes genome-wide) with a threshold of 0.05; and 3) eQTLs were identified for each significant eGene as variants with a locally adjusted p-value from step 1 < 0.1.

To detect QTLs conditional on the lead eQTL, for each gene we repeated the eQTL adding the genotype from the lead eQTL as a covariate (Conditional 1). To detect eQTLs conditional on the top two independent eQTLs (Conditional 2), we repeated eQTL detection using the genotypes from the lead eQTL and the genotype from the lead Conditional 1 eQTL as covariates.

## Enrichment of eQTLs for regulatory elements

To test if eQTLs were enriched for overlapping iPSC regulatory elements, we used the following approach: 1) For all 45,245 tested variants, we determined the overlapping chromatin mark in each of five Roadmap iPSCs; 2) We defined effect size deciles for all tested eQTLs (each eQTL-eGene pair identified by testing 45,245 variants in 146 expressed genes) and binned the eQTLs based on their effect size (*Figure 4—figure supplement 3*, *Supplementary file 6*); 3) For each Roadmap iPSC sample, we counted the number of eQTLs associated with each mark in each bin, creating a 10-by-15 matrix (10 bins and 15 chromatin marks); 4) For normalization, we divided each column by its average across all bins and log2-transformed each value (each chromatin mark was tested individually), in order to obtain enrichments centered around zero; and then 5) we plotted these enrichment values for each chromatin mark (*Figure 4—figure supplement 3*).

## Associations between eQTLs and allele-specific expression in eGenes

We obtained allele-specific expression (ASE) data from our previous genome-wide eQTL analysis on 215 iPSCORE iPSCs (*DeBoever et al., 2017*) and defined the strength of ASE for each gene as the fraction of RNA transcripts that were estimated to originate from the allele with higher expression (hereto referred to as 'allelic imbalance fraction', AIF). We tested the distributions of AIF calculated using MBASED (*Mayba et al., 2014*) on all samples and performed two distinct analyses. First, we tested if the 56 eGenes were more likely to have ASE by comparing their AIF in the samples where their associated lead eQTL was heterozygous with the 90 expressed MHC genes that did not have eQTLs using Mann-Whitney U test (*Figure 4—figure supplement 4*). Next, we tested if eGenes in samples with a heterozygous lead eQTL had stronger AIF than samples homozygous for the lead eQTL using Mann-Whitney U test (*Figure 4—figure supplement 5A*). Finally, we examined all primary and conditional eQTLs (39,891 eQTL-eGene pairs tested, corresponding to 21,739 variants) to determine if eGenes in samples with heterozygous primary eQTLs had stronger AIF than samples homozygous for primary eQTLs (*Figure 4—figure supplement 5B*).

## Associations between eQTLs and GWAS

To test the associations between eQTL signal and complex traits and disease, we downloaded the summary statistics of 4,083 GWAS experiments from the UK BioBank (http://www.nealelab.is/uk-bio-bank). We identified 40,998 variants that were both tested for eQTLs and tested in the UK BioBank. For each of the 56 eGenes and each GWAS experiment, we extracted all p-values associated with these variants and performed colocalization using the coloc.abf from the coloc package in R (*Giambartolomei et al., 2014*). Using this tool, for each tested eGene-GWAS trait pair, we obtained posterior probabilities (PP) for five hypotheses: 1) hypothesis 0 (H0): neither the eGene nor the GWAS have a genetic association in the tested region; 2) H1: only the eGene has a significant association in the tested region; 3) H2: only the GWAS trait has a significant association in the region; 4) H3: both the eGene and the GWAS trait have significant associations, but their causal variants are different; and 5) H4: both the eGene and the GWAS trait and share the same causal variant. The sum of all five posterior probabilities is one for each tested eGene-GWAS trait pair. All pairs with PP H4 >0.2 are reported in *Supplementary file 8*.

## HLA-type eQTL detection

For each sample, the HLA types were assigned dosage values and converted to VCF file format. Dosage was assigned as follows: 0 = the sample did not harbor the analyzed allele; 0.5 = the sample was heterozygous for the analyzed allele; 1 = the sample was homozygous for the analyzed allele. We adjusted the expression levels of all 146 expressed using the same covariates described for the eQTL detection (sex, age, batch, iPSC passage, read depth, read length, ten PEER factors and ten principal components calculated on the genotypes of the 90,081 common SNPs in linkage equilibrium) and performed HLA-type eQTL analysis investigating associations between each single HLA type with allele frequency >1% (283 in total) and expression levels of each of the 146 expressed genes using a linear mixed model (lmekin from the coxme package in R) with a kinship matrix calculated on the 90,081 common SNPs in linkage equilibrium. P-values were adjusted for FDR by using the p.adjust function in R with method = 'BH' (Benjamini-Hochberg) on all combinations between the 146 expressed genes by 283 HLA types present in the 361 individuals (corresponding to 41,318

tests) with a threshold of 0.05. The 15 HLA types that were not in HWE (p<$1 \times 10^{-6}$, *Figure 2—figure supplement 1*) are flagged in *Supplementary file 9*. We initially analyzed self-associations (defined as HLA-type eQTL between an HLA type and the expression of its associated gene, for example HLA*01:01:01:01 and the expression of *HLA-A*) independently to examine associations between expression levels of each HLA gene and its HLA types. To determine if HLA-type eQTLs were independent from single-variant eQTLs, we repeated the HLA-type eQTL detection by conditioning on: 1) the lead primary eQTL (defined as 'conditional 1'); 2) the lead primary and lead conditional 1 eQTLs ('conditional 2'); and 3) on the lead primary, conditional 1 and conditional two lead eQTLs ('conditional 3').

## Coexpression analysis

To test if the four clusters of HLA-type eQTLs (*Figure 6*) identified coexpressed genes associated with the same regulatory variants, we calculated the correlation between the expression levels of each pair of the 146 genes in the MHC region (*Figure 7—figure supplement 1*).

To determine coexpression between each pair of genes in the MHC region, we calculated a linear model as:

$$e_i \sim e_j + covariates$$

Where $e_i$ and $e_j$ represent the expression of the *i-th* and *j-th* genes in the MHC region and covariates are the same used for the eQTL analysis. Beta was used as measure of coexpression.

## Accession numbers

Whole-genome sequencing data of 273 individuals in the iPSCORE cohort (*Panopoulos et al., 2017*) is publicly available through dbGaP: phs001325. RNA-seq data of 215 iPSCs in the iPSCORE cohort (*DeBoever et al., 2017*) is publicly available through dbGaP:phs000924. Whole-genome sequencing data of 377 individuals in the HipSci cohort is publicly available through EGA: PRJEB15299. RNA-seq data of 231 iPSCs in the HipSci cohort is publicly available through ENA: PRJEB7388. eQTL and HLA-type eQTL summary statistics are available at Figshare (https://doi.org/10.6084/m9.figshare.10084733.v1 and https://doi.org/10.6084/m9.figshare.10084736.v1).

## Acknowledgements

We thank Ivan Carcamo-Orive and the other members of the i2QTL Consortium for their comments. This work was supported in part by a California Institute for Regenerative Medicine grant GC1R-06673-B, National Science Foundation CMMI division award 1728497, and NIH grants HG008118, DK105541 and DK112155. MKRD was supported by T15LM011271. These funding agencies played no role in the design or conclusions of this study. The iPSCORE iPSC lines are available; please contact Dr. Kelly A Frazer.

## Additional information

### Funding

| Funder | Grant reference number | Author |
| --- | --- | --- |
| California Institute for Regenerative Medicine | GC1R-06673 | Kelly A Frazer |
| National Institutes of Health | HG008118 | Kelly A Frazer |
| National Institutes of Health | HL107442 | Kelly A Frazer |
| National Institutes of Health | DK105541 | Kelly A Frazer |
| National Institutes of Health | DK112155 | Kelly A Frazer |
| National Science Foundation | CMMI division award 1728497 | Kelly A Frazer |
| National Institutes of Health | T15LM011271 | Margaret KR Donovan |

The funders had no role in study design, data collection and interpretation, or the decision to submit the work for publication.

## Author contributions
Matteo D'Antonio, Formal analysis, Investigation, Methodology, Writing—original draft; Joaquin Reyna, Data curation, Software, Methodology, Writing—original draft; David Jakubosky, Margaret KR Donovan, Hiroko Matsui, Data curation; Marc-Jan Bonder, Resources, Methodology; Oliver Stegle, Supervision; Naoki Nariai, Conceptualization, Data curation, Formal analysis, Methodology; Agnieszka D'Antonio-Chronowska, Resources; Kelly A Frazer, Conceptualization, Supervision, Funding acquisition, Investigation, Writing—original draft, Project administration, Writing—review and editing

## Author ORCIDs
Matteo D'Antonio (iD) https://orcid.org/0000-0001-5844-6433
Kelly A Frazer (iD) https://orcid.org/0000-0002-6060-8902

## Decision letter and Author response
Decision letter https://doi.org/10.7554/eLife.48476.sa1
Author response https://doi.org/10.7554/eLife.48476.sa2

## Additional files

### Supplementary files
• Supplementary file 1. HLA alleles in the 273 iPSCORE subjects and 146 HipSci individuals.(A) HLA types for each iPSCORE individual which includes a row for each gene and a column for the first and second allele. Individuals may be heterozygous, homozygous, or undetermined for one or both alleles. Refer to the Materials and methods section for more details. (B) HLA types for 146 HipSci individuals. For each individual, HLA types derived from both fibroblasts and iPSCs are reported. Individuals with multiple iPSC clones are shown on multiple rows (one row per iPSC clone).

• Supplementary file 2. Unique identifiers for subjects, whole genome sequence and RNA-seq data. The Supplementary File hows for the 446 iPSC samples used in the gene expression and eQTL analyses, the UUID identifiers for iPSCORE data and unique identifiers for HipSci data (subject ID, WGS ID and RNA-seq ID).

• Supplementary file 3. Description of iPSCORE study and HipSci study individuals'.(A) Metadata for subjects in the iPSCORE resource which includes family information, sex, age at enrollment, ethnicity and whether an iPSC clone derived from the subject was used for gene expression analyses. Family information includes a family identifier as well as mother, father, trio and monozygotic twin of which the latter can be used to subset trios and twins respectively. The mother and father columns contain a unique identifier for subjects which participated in the study, empty if parents did not undergo WGS. Ethnicity information is reported in three distinct ways: 1) self-reported information; 2) self-reported information translated into one of seven groups (African American, Asian, European, Hispanic, Indian, Middle Eastern, and Multiple ethnicities reported); and 3) expected superpopulation (SP) from the 1000 Genomes Project. Individuals with iPSCs used in the gene expression analyses are indicated. (B) Metadata for HipSci subjects, including unique subject ID, sex, age at enrollment and ethnicity. All HipSci individuals were unrelated.

• Supplementary file 4. Expression levels of HLA genes.Matrix showing TPM for each HLA gene (columns) in each of the 446 iPSC samples (rows). TPM values are shown for expression estimated obtained by aligning RNA-seq transcripts to (A) reference cDNA sequences and (B) HLA type-specific cDNA sequences. Row names include RNA-seq unique identifiers (*Supplementary file 2*).

• Supplementary file 5. Expression differences between HLA types.For each gene, HLA type, the number of alleles across the 361 individuals with iPSCs, and mean expression (TPM) using HLA type-specific estimates. The p-value (t-test) was calculated by comparing the expression of the HLA type indicated in column B with the expression levels of all the other HLA types for the same cognate

gene. The FDR (Benjamini-Hochberg) is shown in column F. All associations with FDR-adjusted p-value<0.05 were considered as significant.

• Supplementary file 6. eQTLs. The table shows: (A) all the significant eQTLs and (B) the lead eQTL for each gene. eGene information, including gene ID, gene name and TSS position, is shown in columns (A-C). For each variant (position, reference allele, alternative allele, allele frequency and RS ID are shown in columns D-H), eQTL type (primary, conditional 1 or conditional 2), effect size (beta), standard error of beta, p-value and Bonferroni correction are provided (columns I-M). For the 56 eGenes with significant eQTLs at genome-wide level, we observed 76,809 eQTLs (72,473 primary, 2424 conditional 1 and 1912 conditional 2, Bonferroni-adjusted p-value<0.1, column M), corresponding to 27,200 distinct variants.

• Supplementary file 7. Enrichment of eGenes for ASE. The table shows, for 39,891 eQTL-eGene pairs corresponding to 21,739 variants, the number of samples with ASE data in the eGene (i.e. the eGene has at least one heterozygous variant with sufficient sequencing coverage; column D), the number of samples heterozygous for the eQTL variant (column E), the mean allelic imbalance fraction (AIF) in the samples where the eQTL variant is heterozygous (column F) or homozygous (column G), the nominal p-value for the difference in AIF between heterozygous and homozygous samples (Mann-Whitney U test, column H) and Benjamini-Hochberg-adjusted p-value (column I). AIF was calculated using MBASED (*Mayba et al., 2014*) on 215 iPSCORE iPSCs (*DeBoever et al., 2017*).

• Supplementary file 8. Colocalization between eGenes and complex traits and disease. Colocalization between the eQTLs for each eGene and 4,083 UKBB traits with posterior probability of association > 0.5 are shown. For each eGene-trait colocalization, the posterior probabilities of association for all hypotheses (0 = no associations; 1 = signal only from eQTLs; 3 = signal only from GWAS; 3 = different signals from eQTLs and GWAS; and 4 = same signal). Column K lists HLA-genes that have HLA-type eQTLs associated with the eGene in Column B. Column L lists all the HLA-type eQTLs associated with the eGene in Column B.

• Supplementary file 9. eQTL analysis showing associations between HLA types and gene expression. (A) The table shows associations between the expression of each HLA gene and its HLA types (self-associations, eGene = HLA gene). For each association between eGene (columns A-B) and HLA allele (columns C-G), we show whether the HLA type was flagged because it did not pass the HWE threshold (column E, p>$1\times10^{-6}$, *Figure 2—figure supplement 1*), the primary HLA-type eQTL (Type = 'primary') and HLA-type eQTL conditional on the lead eQTL, the top two independent eQTLs and the top three independent eQTLs. Beta, SE of beta, p-value and Benjamini-Hochberg correction are provided for each HLA-type eQTL (primary and conditional). (B) The table shows all non-self HLA-type eQTLs.

• Supplementary file 10. Intersecting eGenes from eQTL and HLA-type eQTL analyses. The table shows, for 118 eGenes (56 with eQTLs, 114 with HLA-type eQTLs, 52 over-lapping) the number of single-variant eQTLs (primary and conditional eQTLs), and the number of HLA-type eQTLs (primary and conditional).

• Transparent reporting form

## Data availability

Whole-genome sequencing data of 273 individuals in the iPSCORE cohort (Panopoulos et al., 2017) is publicly available through dbGaP: phs001325. RNA-seq data of 215 iPSCs in the iPSCORE cohort (DeBoever et al., 2017) is publicly available through dbGaP:phs000924. Whole-genome sequencing data of 377 individuals in the HipSci cohort is publicly available through EGA: PRJEB15299. RNA-seq data of 231 iPSCs in the HipSci cohort is publicly available through ENA: PRJEB7388. eQTL and HLA-type eQTL summary statistics are available at Figshare (https://doi.org/10.6084/m9.figshare.10084733.v1 and https://doi.org/10.6084/m9.figshare.10084736.v1).

The following datasets were generated:

| Author(s) | Year | Dataset title | Dataset URL | Database and Identifier |
|---|---|---|---|---|
| Matteo D'Antonio, Kelly A Frazer | 2019 | HLA-type eQTLs | https://doi.org/10.6084/m9.figshare.10084733.v1 | figshare, 10.6084/m9.figshare.10084733.v1 |

| | | | | |
|---|---|---|---|---|
| Matteo D'Antonio, Kelly A Frazer | 2019 | eQTLs in the MHC region | https://doi.org/10.6084/m9.figshare.10084736.v1 | figshare, 10.6084/m9.figshare.10084736.v1 |

The following previously published datasets were used:

| Author(s) | Year | Dataset title | Dataset URL | Database and Identifier |
|---|---|---|---|---|
| DeBoever C, Li H, Jakubosky D, Benaglio P, Reyna J, Olson KM, Huang H, Biggs W, Sandoval E, D'Antonio M | 2017 | NextGen Consortium: The iPSCORE (iPSC Collection for Omic Research) Resource for studying the impact of genetic variation on molecular and physiological phenotypes | https://www.ncbi.nlm.nih.gov/projects/gap/cgi-bin/study.cgi?study_id=phs000924.v1.p1 | dbGaP, phs000924 |
| Panopoulos AD, D'Antonio M, Benaglio P, Williams R, Hashem SI, Schuldt BM, DeBoever C, Arias AD, Garcia M, Nelson BC | 2017 | NextGen Consortium: The iPSCORE (iPSC Collection for Omic Research) Resource for studying the impact of genetic variation on molecular and physiological phenotypes - Whole Genome Sequence | https://www.ncbi.nlm.nih.gov/projects/gap/cgi-bin/study.cgi?study_id=phs001325.v1.p1 | dbGaP, phs001325 |
| Wellcome Sanger Institute | 2016 | HipSci Whole genome sequencing healthy volunteers | https://www.ebi.ac.uk/ena/data/view/PRJEB15299 | European Nucleotide Archive, PRJEB15299 |
| Wellcome Sanger Institute | 2016 | HipSci RNAseq healthy volunteers | https://www.ebi.ac.uk/ena/data/view/PRJEB7388 | European Nucleotide Archive, PRJEB7388 |

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
