## [Decision Letter]

**Acceptance summary:**

The MHC is the region of the human genome associated with the greatest number of different human diseases. Despite this, unravelling the relationship between genetic variation in the region and the expression of HLA and non-HLA genes with relevance to disease is a significant challenge. Here the authors have used deep sequencing of stem cells from 419 individuals to create an extensive map of regulatory variation in the MHC region, exploring genetic associations for over 4,000 different disease and traits, and uncovering links between 180 such associations with variation in the expression of specific genes. This study provides insights into the genetic architecture of the MHC region and sheds new light on the relatively underappreciated concept that non-coding, regulatory genetic variation in the MHC region can also contribute to disease risk. Therefore, this work has important implications for understanding the role of the MHC in human disease, and will be of interest to geneticists, immunologists, molecular biologists and clinical scientists.

**Decision letter after peer review:**

Thank you for submitting your article "In-depth genetic analysis of 6p21.3 reveals insights into associations between HLA types and complex traits and disease" for consideration by *eLife*. Your article has been reviewed by three peer reviewers, and the evaluation has been overseen by a Reviewing Editor and Mark McCarthy as the Senior Editor. The following individuals involved in review of your submission have agreed to reveal their identity: Vivian G Cheung (Reviewer #1); Heather Cordell (Reviewer #2).

The reviewers have discussed the reviews with one another and the Reviewing Editor has drafted this decision to help you prepare a revised submission.

Summary:

In this manuscript whole genome sequencing is employed to obtain HLA genotypes, which are then used to in a gene expression analysis to identify the regulatory variants affecting HLA and non-HLA genes in the major histocompatibility complex region of human chromosome 6. The study implies that HLA types can predict the expression of non-HLA genes, and identifies potential clusters of co-expressed genes. As the HLA region is highly polymorphic and many of its variants have been associated with immune-based and other human diseases, an accurate approach for determining HLA genotypes from whole genome sequencing data, and an improved understanding of the impact of variants within the region on modulating gene expression, are important.

The essential revisions detailed below provide the collated comments and queries from the reviewers that need to be addressed. The reviewers were in agreement with respect to the need to give the paper a more cohesive structure, defining one or a few key questions and focusing on the most novel findings. We think that the suggestions regarding the statistical methods and the employment of simulation-based approaches are helpful and should be carefully considered. All reviewers also commented on the need to provide further support for the eQTL findings. Although we do not consider additional experimental validation to be necessary, we would like an indication of whether the eQTLs found were shared with those reported in previous studies or other available data sets. Other types of evidence that can support the eQTLs, such as the localisation of relevant variants in regulatory elements or GWAS enrichment of the variants, should also be investigated and reported.

Essential revisions:

1) Ideas presented in the Introduction are not taken up in the conclusion. The Introduction refers to "altered self" and "mistaken identity" hypotheses. Although the latter is taken up in the proof of principle, these terms are not used again, making the Introduction and Discussion somewhat disjointed. More generally, the paper covers a broad array of issues, but lacks focus: it was unclear exactly what the main idea is. If the focus is on demonstrating the enrichment of sites with regulatory role, a concern is that with respect to comparisons across studies with power that is non-comparable. If the idea is to defend the "mistaken identity" hypothesis, this needs to be taken up more thoroughly in the Discussion, given that the idea is now built upon, almost exclusively, a case study of one locus and its putative HLA-eQTL.

2) The history of HLA typing need to be amended. The text states that "the development of array-based technologies provided methods to estimate HLA types at four-digit resolution". Four-digit resolution greatly predates array technologies, which were brought in late in the history of HLA typing as a means to impute four-digit alleles, given appropriate reference samples and SNP density. Similarly, the possibility of "incorporating synonymous variants and non-coding regulatory variants" is attributed to NGS-based methods (supported by a string of citations between 2012 and 2017) but (Sanger) sequencing and PCR-based with allele-specific primers describing these levels of variation have long been around (in fact, the data in the IMGT resource was generated by Sanger sequencing of cloned sequences). The critical step of the cited papers was in fact adapting NGS pipelines for HLA typing.

3) In Figure 4, there are 3 groups of genes (described in the subsection “Genes in the MHC region are enriched for SNP-eQTLs”): those who expression levels show association with 283 SNP alleles, those with 10 or fewer and those with more than 1,000 associations. The data shown in Figure 4 do not show the distributions of expression levels for each allele so it is not possible to tell whether the associations are driven by just a few samples. Can the authors replot some of the results with graphs where expression levels are on the y-axis, and the three genotype groups on the x-axis?

4) An important aspect of the study is the identification of "four groups of genes that have alleles with highly correlated expression". This is largely supported by the results in Figure 5, which show sets of genes with significant correlations with HLA genotype (i.e., HLA-eQTL in the paper's terminology). This is a potentially interesting finding. Why was this analysis carried out in the first place? What are prior results that motivate the expectation that HLA allele identity will be predictive of expression of neighboring genes? Is this a causal mechanism or are HLA genes marking a haplotype with other genetic signatures? If documenting co-expression is of interest, why not examine it directly, rather than mediated through associations to an HLA type?

5) In the treatment of HLA-eQTLs, an interesting idea that yielded useful results, the authors chose not to examine the role of HLA genotypes on the expression of other HLA genes in the MHC region. This examination should either be expanded upon or a justification for the lack of a more extensive analysis needs to be provided.

6) The entire discussion of how levels of variability at 2 and 8 digit relate occupies a fair portion of the paper. Although essentially correct, much of the pattern seen is naturally expected given the nesting of 8 digit definitions within 2 digits. Thus, large sections of the paper relating diversity, using correlation analyses (Figure 2) and discussions of findings could be omitted. The very definition of these levels makes the patterns found and described expected. This shows up again in the discussion of concordance, where the fact that the values are higher for 2 digits is presented a result. This is unavoidable, given the nesting, which constrain the results so that the 2-digit concordance can only be the same as or greater than that at 8-digits. This is a feature of the data, not a result.

7) The paper dedicates considerable space to discussing variation among alleles. The usage of "allele-specific expression" as employed here does not match the usual definition, which refers to the "comparing the expression of the alleles in a heterozygous individual (allele specific expression or ASE analysis)" (https://doi.org/10.1186/s12864-018-5181-0). What is being done is in fact a test for differences in expression among alleles, a different question. Figure 3 treats the data as "each point reporting one allele". However, in the text referring to this same analysis we read "We (…) compared the TPM distributions between all samples carrying a specific HLA allele and all the other samples." This is a different analysis, since the two alleles of heterozygous individuals are confounded, both contributing to the locus-level expression. Although sometimes employed, extrapolating an individual's expression to an allele which is carried is an approach that introduces an additional source of uncertainty, which can be dealt with by analyzing the allele-level expression (since the allelic sequence in present in the index, allele-level expression could be directly compared). The paper needs to clarify what was done and justify the choices.

8) Throughout the manuscript, the authors refer to the personalized component in the index used to estimate expression levels as "personalized transcripts". However, according to the Methods, cDNA sequences from the IMGT database were used instead of personalized isoform sequences. While this is a personalized index at the HLA level, it is not a personalized transcriptome-based index since transcriptomic diversity is not being assayed as in Gensterblum-Miller et al. (https://doi. org/10.4049/jimmunol.1701061).

9) The paper fails to acknowledge previous work. The statement "The inability to genetically interrogate the MHC region" fails to acknowledge work such as Lee et al. (https://doi.org/10.1093/bioinformatics/bty), Jensen et al. (10.1101/gr.218891.116), Aguiar et al. (10.1371/journal.pgen.1008091). Similarly, the paper states that high resolution data for high depth data has not been provided across the region, but the above papers do address this issue. Later, in Results, when discussing the use of personalized transcripts to determine gene expression levels the authors argue that "the expression levels of genes in this region measured by RNA-seq may be inaccurate due to large numbers of mismatches between sequence reads and reference sequence" but do not have results on this, and fail to cite papers such as Aguiar et al., 2019, which explores exactly the relationship between mismatches and accuracy of expression through simulations or more general treatments such as Panousis et al., 2014.

10) The paper presents several analyses to support the reliability of the HLA genotype calls. Initially, the study examines the SNP density interferes with alignment coverage. It then argues that values which are sequencing depths (not coverage) are comparable to other genome regions and thus claims this implies "high quality". While the info on coverage and depth is useful, it is not clear that these values alone can be treated as proxies of high quality.

11) The paper also uses concordance among twins to "examine the accuracy of the high-resolution HLA types". While concordance is useful to document if there are no experimental errors, it does not guarantee accuracy. For example, if there is a systematic error in the alignment procedure, the same and incorrect sample could be obtained in each case. Thus, this may be indicated as an assay to measure experimental replicability, but not accuracy.

12) The argument for an improvement in expression estimates using the personalized pipeline (subsection “Using personalized transcripts to determine gene expression levels”) is that class I estimates are higher and class II are lower, when the pipeline is used. It is unclear why this is support for greater accuracy. It would imply that the higher expression of class II with the standard reference index results in overestimates, but mismatches between that index and the individual's alleles don't make such a prediction. The fact that HLA class II genes have lower expression doesn't naturally lead to the prediction that the effect of a better index would be to lower expression. Accuracy would require some external measure of expression (which isn't available) or a simulation-based approach (as implemented in some previously cited studies).

13) The paper spends some time correlating genotype and allele frequencies, including a report of negative correlation between number of alleles and number of homozygous genotypes. The standard way to explore this is to test for deviations from Hardy-Weinberg proportions, something that can be done using various existing software packages.

14) The study also shows a correlation between allele frequencies in the two datasets. While this is comforting, it is a very coarse indicator, which at most could be reported as a supplementary result, but doesn't warrant inclusion in the body of the paper.

15) For the eQTL mapping, the authors have adjusted the expression levels for different batches, ethnicities etc, but have not controlled for the effects of population structure on the genotype data (see for example the use of PCA in Lappalainen et al., Nature, 2013). It would also be helpful to provide more details on how the samples were used in the analyses. More details on how the adjustments were made to ensure that ethnicity did not add unexpected structure to the data would be helpful. Additionally, only one of the MZ twins should be used – it is not clear that the twins were removed.

16) The authors should justify the use of the overly conservative Bonferroni correction in the eQTL mapping instead of the widely-used FDR correction in genome-wide studies. By querying the MHC region alone, the definition of significance becomes different from genome-wide studies, making contrasts between these (including the proportion of eGenes) not directly comparable. Wouldn't it be preferable to carry out a genome-wide analysis, and then use the distribution of p-values to estimate corrected values under a chosen FDR? This may give readers a better ability to place these findings in the context of previous eQTL studies.

17) The authors claim that there is an enrichment of eGenes on the MHC region by comparing with a previous genome-wide work which analyzed only a subset of the samples (215 of the 446 samples). Since sample size affects power to detect eGenes, the direct comparison is not appropriate. More generally, the notion that "in the MHC region a large fraction of expressed genes are eGenes" should be stated in that qualifies the identify of an eGene as contingent on the dataset. A gene can be an eGene in some studies and not others, as a function of sampling. Thus, the statement that "genes in the MHC region are significantly enriched for having their expression levels modulated by regulatory variants" is not supported by the analysis.

18) Further support for the eQTL findings. For some of the regulatory variants, are there any samples that are heterozygous and show that the two alleles are indeed expressed at the different levels? Were the eQTL variants enriched for localisation in regulatory sites? Were they enriched for GWAS signals and have the eGenes in question been previous linked to disease?

---

## [Author Response]

Essential revisions:1) Ideas presented in the Introduction are not taken up in the conclusion. The Introduction refers to "altered self" and "mistaken identity" hypotheses. Although the latter is taken up in the proof of principle, these terms are not used again, making the Introduction and Discussion somewhat disjointed. More generally, the paper covers a broad array of issues, but lacks focus: it was unclear exactly what the main idea is. If the focus is on demonstrating the enrichment of sites with regulatory role, a concern is that with respect to comparisons across studies with power that is non-comparable. If the idea is to defend the "mistaken identity" hypothesis, this needs to be taken up more thoroughly in the Discussion, given that the idea is now built upon, almost exclusively, a case study of one locus and its putative HLA-eQTL.

We thank the reviewers for their thoughtful and insightful comments and agree that the manuscript lacked focus. For these reasons, we have substantially re-written the manuscript and have focused on the following topics:

• Using whole genome sequence from 419 individuals to call eight-digit HLA types and RNA-seq data from 361 matched induced pluripotent stem cells (iPSCs) to create a comprehensive map of regulatory variants in the MHC region.

• Building on this regulatory map, we explore GWAS signals for 4,083 complex traits, and finding strong evidence of colocalization for 180 disease loci with expression quantitative trait loci (eQTLs).

• Showing that taking eight-digit HLA type haplotypes into account substantially increases power to identify eGenes over single-variant eQTL analysis.

• Examining the association between the 8.1 ancestral haplotype and delayed colonization in Cystic Fibrosis, and postulating that downregulation of *RNF5* expression is the likely causal mechanism.

2) The history of HLA typing need to be amended. The text states that "the development of array-based technologies provided methods to estimate HLA types at four-digit resolution". Four-digit resolution greatly predates array technologies, which were brought in late in the history of HLA typing as a means to impute four-digit alleles, given appropriate reference samples and SNP density. Similarly, the possibility of "incorporating synonymous variants and non-coding regulatory variants" is attributed to NGS-based methods (supported by a string of citations between 2012 and 2017) but (Sanger) sequencing and PCR-based with allele-specific primers describing these levels of variation have long been around (in fact, the data in the IMGT resource was generated by Sanger sequencing of cloned sequences). The critical step of the cited papers was in fact adapting NGS pipelines for HLA typing.

To focus the manuscript and Introduction, we have removed this entire section describing the different resolutions of HLA typing. Instead we have added a new figure (Figure 1) that describes current standard HLA nomenclature

3) In Figure 4, there are 3 groups of genes (described in the subsection “Genes in the MHC region are enriched for SNP-eQTLs”): those who expression levels show association with 283 SNP alleles, those with 10 or fewer and those with more than 1,000 associations. The data shown in Figure 4 do not show the distributions of expression levels for each allele so it is not possible to tell whether the associations are driven by just a few samples. Can the authors replot some of the results with graphs where expression levels are on the y-axis, and the three genotype groups on the x-axis?

We thank the reviewers for this comment: to show that the eQTL associations are not driven by just a few samples we added Figure 4—figure supplement 2, which shows boxplots of the expression levels for each of the 56 eGenes divided by the three genotypes (homozygous reference, heterozygous and homozygous alternative) of the lead eQTL. These plots show that the eQTL signals are supported by many samples. We have also made the eQTL summary statistics publicly available

(https://figshare.com/s/c8533530204e1822c6f4 and https://figshare.com/s/eab712af0e4baeb99251).

4) An important aspect of the study is the identification of "four groups of genes that have alleles with highly correlated expression". This is largely supported by the results in Figure 5, which show sets of genes with significant correlations with HLA genotype (i.e., HLA-eQTL in the paper's terminology). This is a potentially interesting finding. Why was this analysis carried out in the first place? What are prior results that motivate the expectation that HLA allele identity will be predictive of expression of neighboring genes? Is this a causal mechanism or are HLA genes marking a haplotype with other genetic signatures? If documenting co-expression is of interest, why not examine it directly, rather than mediated through associations to an HLA type?

We thank the reviewers for this observations. We agree that the rationale for analyzing the four sets of genes with significant associations with HLA genotypes was not clearly explained.

Why was this analysis carried out in the first place?

We hypothesized that HLA types, which represent a haplotype of many regulatory and coding SNPs, could exert a greater effect on gene expression than single variants. This is now clearly stated in the Discussion and alluded to in the Results.

What are prior results that motivate the expectation that HLA allele identity will be predictive of expression of neighboring genes?

There were no prior results that motived us to hypothesize that this would be the case. However, we have since identified a previous study that conducted a similar analysis but on a considerably smaller scale. We have cited a recent paper by Lam et al.(Lam et al., 2017), which describes eQTLs in the MHC region in 31 LCL samples, in the Discussion.

Is this a causal mechanism or are HLA genes marking a haplotype with other genetic signatures?

We believe that these findings suggest that the HLA-type eQTLs capture regulatory variants associated with the expression of one or a few neighboring genes. Furthermore, unlike the eQTLs from the single-variant analysis that tend to span long distances most likely due to the strong LD in the MHC region, the HLA-type eQTLs capture regulatory variants that act over relatively short distances.

We have expanded the Results and Discussion to describe these associations in detail.

If documenting co-expression is of interest, why not examine it directly, rather than mediated through associations to an HLA type?

In response to this comment, we have added a new analysis section in the Results “HLA-type eQTLs capture regulatory variation associated with expression of neighboring genes” and in the Materials and methods section “Coexpression analysis”.

“We investigated the molecular underpinnings of the clustering of the most significant

HLA-type eQTLs within the four distinct genomic regions (Figure 6A): 1)

chr6:29640168-30152231, including HLA class I genes (*HLA-A, HLA-F* and *HLA-G*) and HLA class I pseudogenes; 2) chr6:31171745-31512238, including HLA class I genes

(*HLA-B, HLA-C), MICA* and *MICB*; 3) chr6: 32361463-32789609, including HLA class

II genes (*HLA-DRA, HLA-DRB1, HLA-DQA1, HLA-DQB1* and *HLA-DOB), TAP1* and

*TAP2*; and 4) chr6:32916389-33115544, including HLA class II genes (*HLA-DPA1, HLA-DPB1*, and *HLA-DPB2*). We observed that different eGenes in each cluster were typically associated with the genotypes of different HLA types, suggesting that the observed clustering was not due to co-expression (i.e., correlated expression levels) of neighboring HLA genes (Figure 7). There was one exception in cluster 3, with the expression of both *HLA-DRB5* and *HLA-DRB1* associated with many of the same HLA types (Figure 7C). For this reason, we examined the co-expression of the eGenes in each of the four clusters. Only five eGenes in cluster 3 (*HLA-DRA, HLA-DRB5, HLA-DQA1, HLA-DQB1* and *HLA-DRB1*) displayed moderate correlated expression (Figure 7—figure supplement 1), despite having their expression levels largely associated with different HLA types of neighboring genes (Figure 7). These findings suggest that the HLA-type eQTLs capture regulatory variants associated with the expression of one or a few neighboring genes. Furthermore, unlike the eQTLs from the single-variant analysis that tend to span long distances most likely due to the strong LD in the MHC region (Figure 4), the HLA-type eQTLs capture regulatory variants that act over relatively short distances.”

5) In the treatment of HLA-eQTLs, an interesting idea that yielded useful results, the authors chose not to examine the role of HLA genotypes on the expression of other HLA genes in the MHC region. This examination should either be expanded upon or a justification for the lack of a more extensive analysis needs to be provided.

We apologize for the confusion: in our original submission, we only removed “self” associations, i.e., we did not analyze the role of HLA genotypes on the same HLA gene (for example HLAA:01:01:01:01 and *HLA-A* expression), but we investigated the associations between the HLA type of one HLA gene and the expression of other HLA genes (for example HLA-A:01:01:01:01 and *HLA-B* expression). In the new version of the manuscript, we have added the self-association analysis (for example HLA-A:01:01:01:01 and *HLA-A* expression) in the Results section “HLAtype eQTLs associated with the expression of 114 eGenes in the MHC region” and in Supplementary file 9. We have tried to make the text in the Results and Materials and methods describing this analysis clearer.

6) The entire discussion of how levels of variability at 2 and 8 digit relate occupies a fair portion of the paper. Although essentially correct, much of the pattern seen is naturally expected given the nesting of 8 digit definitions within 2 digits. Thus, large sections of the paper relating diversity, using correlation analyses (Figure 2) and discussions of findings could be omitted. The very definition of these levels makes the patterns found and described expected. This shows up again in the discussion of concordance, where the fact that the values are higher for 2 digits is presented a result. This is unavoidable, given the nesting, which constrain the results so that the 2-digit concordance can only be the same as or greater than that at 8-digits. This is a feature of the data, not a result.

We agree that the discussion about the differences between two-digit and eight-digit resolutions is superfluous and contributes to the lack of focus that the reviewers pointed out in Comment 1. Therefore, we have removed this analysis from the manuscript including the Results and Materials and methods, and all related figures.

7) The paper dedicates considerable space to discussing variation among alleles. The usage of "allele-specific expression" as employed here does not match the usual definition, which refers to the "comparing the expression of the alleles in a heterozygous individual (allele specific expression or ASE analysis)" (https://doi.org/10.1186/s12864-018-5181-0). What is being done is in fact a test for differences in expression among alleles, a different question. Figure 3 treats the data as "each point reporting one allele". However, in the text referring to this same analysis we read "We (…) compared the TPM distributions between all samples carrying a specific HLA allele and all the other samples." This is a different analysis, since the two alleles of heterozygous individuals are confounded, both contributing to the locus-level expression. Although sometimes employed, extrapolating an individual's expression to an allele which is carried is an approach that introduces an additional source of uncertainty, which can be dealt with by analyzing the allele-level expression (since the allelic sequence in present in the index, allele-level expression could be directly compared). The paper needs to clarify what was done and justify the choices.

We have greatly reduced the space in the text devoted to discussing expression differences among HLA types (alleles). We have eliminated all discussion of comparing alignments to reference versus cDNA sequences specific for the HLA types from the Results. We moved some of this to the supplementary figures.

We apologize for the confusion in the description of the analysis about obtaining expression levels from alignments to cDNA sequences specific for the HLA types. We have edited the Materials and methods to clarify that we treated homozygous and heterozygous individuals in different ways:

“The IMGT-IPD HLA database release 3.30.0(Robinson et al., 2016), RSEM version 1.2.20(Li and Dewey, 2011) contained cDNA sequences corresponding to each eight-digit resolution allele of the 24 HLA genes expressed in iPSCs. […] For heterozygous individuals, the TPM associated with each cDNA sequence was considered as HLA type-specific expression, while for homozygous individuals, the TPM was divided by two to obtain HLA type-specific expression levels.”

8) Throughout the manuscript, the authors refer to the personalized component in the index used to estimate expression levels as "personalized transcripts". However, according to the Materials and methods, cDNA sequences from the IMGT database were used instead of personalized isoform sequences. While this is a personalized index at the HLA level, it is not a personalized transcriptome-based index since transcriptomic diversity is not being assayed as in Gensterblum-Miller et al. (https://doi. org/10.4049/jimmunol.1701061).

We agree that “personalized transcripts” is not the correct term to use. As discussed in response to comment 7, we have eliminated all discussion of comparing alignments to reference versus cDNA sequences specific for the HLA types from the Results. We referenced the paper by Gensterblum-Miller et al. in the Results section “Eight-digit HLA types associated with expression of cognate HLA gene”

9) The paper fails to acknowledge previous work. The statement "The inability to genetically interrogate the MHC region" fails to acknowledge work such as Lee et al. (https://doi.org/10.1093/bioinformatics/bty), Jensen et al. (10.1101/gr.218891.116), Aguiar et al. (10.1371/journal.pgen.1008091). Similarly, the paper states that high resolution data for high depth data has not been provided across the region, but the above papers do address this issue. Later, in Results, when discussing the use of personalized transcripts to determine gene expression levels the authors argue that "the expression levels of genes in this region measured by RNA-seq may be inaccurate due to large numbers of mismatches between sequence reads and reference sequence" but do not have results on this, and fail to cite papers such as Aguiar et al., 2019, which explores exactly the relationship between mismatches and accuracy of expression through simulations or more general treatments such as Panousis et al., 2014.

We thank the reviewers for these comments. As part of focusing the paper we have removed all sections to which comment 9 is concerning. Nevertheless we have included references for: Jensen et al.(Jensen et al., 2017), Lee et al.(Lee et al., 2018), Panousis et al.(Panousis et al., 2014) and Aguiar et al.(Aguiar et al., 2019).

10) The paper presents several analyses to support the reliability of the HLA genotype calls. Initially, the study examines the SNP density interferes with alignment coverage. It then argues that values which are sequencing depths (not coverage) are comparable to other genome regions and thus claims this implies "high quality". While the info on coverage and depth is useful, it is not clear that these values alone can be treated as proxies of high quality.

In our original manuscript, we had concluded that WGS samples were high-quality because the read depth in the MHC region is comparable to the rest of the genome and that the SNP density is comparable to previous studies. This was based on the assumptions that, if the alignment were low-quality, the read depth would be lower than the rest of the genome and the SNP density distribution would be noisy and would not have the same shape as was previously observed. We understand that these assumptions are not sufficient to claim that WGS had high quality.

To address this comment, we have changed the title of the first section of the Results to “Eight-digit HLA typing” and we removed any reference to quality. We have modified the take-home message of this section to the fact that the alignment of all WGS samples is comparable to the rest of the genome and that the SNP density is similar to previous studies(Norman et al., 2017).

11) The paper also uses concordance among twins to "examine the accuracy of the high-resolution HLA types". While concordance is useful to document if there are no experimental errors, it does not guarantee accuracy. For example, if there is a systematic error in the alignment procedure, the same and incorrect sample could be obtained in each case. Thus, this may be indicated as an assay to measure experimental replicability, but not accuracy.

We agree that concordance is not a measure of accuracy. When we describe high concordance we no longer refer to this as “accuracy” but rather “replicability” and have changed the section title in the Results section to “HLA types have high recall rates and reproducibility”.

12) The argument for an improvement in expression estimates using the personalized pipeline (subsection “Using personalized transcripts to determine gene expression levels”) is that class I estimates are higher and class II are lower, when the pipeline is used. It is unclear why this is support for greater accuracy. It would imply that the higher expression of class II with the standard reference index results in overestimates, but mismatches between that index and the individual's alleles don't make such a prediction. The fact that HLA class II genes have lower expression doesn't naturally lead to the prediction that the effect of a better index would be to lower expression. Accuracy would require some external measure of expression (which isn't available) or a simulation-based approach (as implemented in some previously cited studies).

We have eliminated most of the discussion on personalized expression from the manuscript. See answers to comments 7 and 8 for a description as to how we modified the manuscript.

13) The paper spends some time correlating genotype and allele frequencies, including a report of negative correlation between number of alleles and number of homozygous genotypes. The standard way to explore this is to test for deviations from Hardy-Weinberg proportions, something that can be done using various existing software packages.

We agree with the reviewers and have removed all the analyses describing correlations between genotypes and allele frequencies.

We have added the following to the Results:

“To estimate the accuracy of the 526 called HLA alleles we tested for Hardy-Weinberg equilibrium (HWE) in the unrelated individuals and found that the vast majority (511, 97.1%) were in HWE (p > 1 x 10^-6^, Figure 2—figure supplement 1A-C), consistent with the reproducibility values in twin pairs (96.7%) and in fibroblast-iPSC pairs (95.7%). […] Overall, HLA-VBSeq(Nariai et al., 2015) was able to accurately detect eight-digit resolution HLA alleles in the 650 WGS samples as evidenced by high recall rates, reproducibility and allele frequencies consistent with Hardy-Weinberg law.”

And added the following to the Materials and methods:

“Hardy-Weinberg equilibrium (HWE)

To investigate HWE, we performed a likelihood ratio test on the allele frequency of each HLA allele independently in 275 unrelated Caucasians in the iPSCORE and HipSci cohorts. […] All associations with p-values < 1 x 10^-6^ were considered as significantly deviating from HWE and flagged in the HLA-type QTL analysis.”

14) The study also shows a correlation between allele frequencies in the two datasets. While this is comforting, it is a very coarse indicator, which at most could be reported as a supplementary result, but doesn't warrant inclusion in the body of the paper.

We agree with the reviewers and have removed the figure showing the correlation between allele frequencies in the two datasets from the manuscript.

15) For the eQTL mapping, the authors have adjusted the expression levels for different batches, ethnicities etc, but have not controlled for the effects of population structure on the genotype data (see for example the use of PCA in Lappalainen et al., Nature, 2013). It would also be helpful to provide more details on how the samples were used in the analyses. More details on how the adjustments were made to ensure that ethnicity did not add unexpected structure to the data would be helpful. Additionally, only one of the MZ twins should be used – it is not clear that the twins were removed.

We agree that the use of a linear model (such as MatrixEQTL) to test for eQTLs is not appropriate when tested individuals are related. Therefore, we repeated the eQTL analysis (both on single variants and HLA types) using a linear mixed model that includes a kinship matrix to take population structure into account.

We have substantially re-written the Materials and methods to describe these analyses:

“eQTL detection

For eQTL analyses, we used the WGS from 361 individuals (215 iPSCORE and 146 HipSci, Supplementary file 2) for which iPSC RNA-seq data were available. […] We performed eQTL analysis for the 146 expressed genes and the 45,245 variants using a linear mixed model (lmekin from the coxme package in R) with a kinship matrix (to take population structure into account, including multiple ethnicities, members of the same family and twin pairs) calculated on 90,081 common SNPs distributed across the genome and in linkage equilibrium.”

We have changed Supplementary files 6, 7 and 9 with the new QTL results.

16) The authors should justify the use of the overly conservative Bonferroni correction in the eQTL mapping instead of the widely-used FDR correction in genome-wide studies. By querying the MHC region alone, the definition of significance becomes different from genome-wide studies, making contrasts between these (including the proportion of eGenes) not directly comparable. Wouldn't it be preferable to carry out a genome-wide analysis, and then use the distribution of p-values to estimate corrected values under a chosen FDR? This may give readers a better ability to place these findings in the context of previous eQTL studies.

We thank the reviewers for suggesting how we could improve the eQTL analysis. In response to Comment 15, we have changed the method to perform the eQTL analysis from a linear model to a linear mixed model to account for random genetic effects, which has allowed us to use all the iPSC samples (including multiple samples from the same individual and from twin pairs) and therefore has improved our eQTL detection power.

To address the reviewers’ comments, we improved the explanation of the eQTL detection method in the Materials and methods section “eQTL detection”.

“In order to detect significant eQTLs for each eGene, and then detect eGenes at genomewide significance levels, we employed a three-step procedure to perform hierarchical multiple testing correction (Huang et al., 2018): 1) locally adjusted p-value at gene-level: nominal p-values were Bonferroni-corrected for the number of tests performed for each gene (45,245, Supplementary file 7A); the variant with the minimum adjusted p-value was considered as lead eQTL (Supplementary file 7B); 2) genome-wide adjusted p-value of lead eQTLs: to FDRcorrect across all expressed genes, we further adjusted the p-values of the lead eQTLs using the p.adjust function in R with method = “BH” (Benjamini-Hochberg) and n = 20,595 (corresponding to the number of expressed genes genome-wide) with a threshold of 0.05; and 3) eQTLs were identified for each significant eGene as variants with a locally adjusted p-value from step 1 < 0.1.”

17) The authors claim that there is an enrichment of eGenes on the MHC region by comparing with a previous genome-wide work which analyzed only a subset of the samples (215 of the 446 samples). Since sample size affects power to detect eGenes, the direct comparison is not appropriate. More generally, the notion that "in the MHC region a large fraction of expressed genes are eGenes" should be stated in that qualifies the identify of an eGene as contingent on the dataset. A gene can be an eGene in some studies and not others, as a function of sampling. Thus, the statement that "genes in the MHC region are significantly enriched for having their expression levels modulated by regulatory variants" is not supported by the analysis.

We agree that it would be misleading to compare the number of eQTLs across studies, where many other factors (most importantly, sample size) play important roles. Therefore, we have removed this statement from the text (Introduction, Results and Discussion).

18) Further support for the eQTL findings. For some of the regulatory variants, are there any samples that are heterozygous and show that the two alleles are indeed expressed at the different levels? Were the eQTL variants enriched for localisation in regulatory sites? Were they enriched for GWAS signals and have the eGenes in question been previous linked to disease?

We thank the reviewers for these suggestions. To respond to this comment we performed three additional analyses (explained below).

For some of the regulatory variants, are there any samples that are heterozygous and show that the two alleles are indeed expressed at the different levels?

To respond to this comment, we performed enrichment analyses to determine if eQTLs were associated with allele-specific expression (ASE):

1) We tested if the 56 eGenes were more likely to have ASE than the other expressed genes in the MHC region. We obtained ASE data from our previous eQTL analysis on 215 iPSCORE iPSCs(DeBoever et al., 2017) and tested the distributions of major allele frequencies calculated using MBASED(Mayba et al., 2014) on all samples between the 56 eGenes and the other 90 genes expressed in the MHC region. We found that eGenes were significantly more likely to have ASE [mean allelic imbalance fraction (AIF) = 67.3%, compared with 61.9% in the other genes, p = 4.38 x 10^-64^, Mann-Whitney U test, Figure 4—figure supplement 4, Supplementary file 7].

2) Next, we investigated if eGenes in samples with a heterozygous lead eQTL had stronger AIF than samples homozygous for the eQTL using Mann-Whitney U test. We observed that eGenes in samples with heterozygous eQTLs (primary and conditional) have stronger ASE compared with eGenes in samples homozygous for eQTLs (39,891 eQTLs in total examined, corresponding to 21,739 variants; Figure 4—figure supplement 5A, B).

We describe these analyses in the Results (“eQTLs associated with the expression of 56 eGenes in the MHC region”), in the Materials and methods (“Associations between eQTLs and allele-specific expression in eGenes”), Figure 4—figure supplements 4 and 5.

Were the eQTL variants enriched for localisation in regulatory sites?

We obtained 15 ChromHMM(Ernst and Kellis, 2012) chromatin states in five iPSC lines from the Roadmap Epigenomics Consortium(Roadmap Epigenomics et al., 2015) and tested if the effect size of eQTLs changed between states. Intuitively, variants in regulatory elements are more likely to be associated with gene expression, therefore have stronger effect size. To perform this analysis, we used the following approach:

1) For all 45,245 tested variants, we determined the overlapping chromatin mark in each of five Roadmap iPSCs;

2) We defined effect size deciles for all tested eQTLs (each eQTL-eGene pair identified by testing 45,245 variants in 146 expressed genes) and binned the eQTLs based on their effect size (Figure 4—figure supplement 3, Supplementary file 6);

3) For each Roadmap iPSC sample, we counted the number of eQTLs associated with each mark in each bin, creating a 10-by-15 matrix (10 bins and 15 chromatin marks);

4) For normalization, we divided each column by its average across all bins and log2-transformed each value (each chromatin mark was tested individually), in order to obtain enrichments centered around zero;

5) We plotted these enrichment values for each chromatin mark (Figure 4—figure supplement 3).

Figure 4—figure supplement 3 shows that, in all Roadmap iPSC samples, the eQTLs in the MHC region with the largest effect sizes were enriched for overlapping regulatory elements active in iPSCs including active transcription start sites (TSS), regions flanking active TSS, UTRs (defined as regions transcribed at 5’ and 3’) and transcribed regions.

We added a description of this analysis in the Results section “eQTLs associated with the expression of 56 eGenes in the MHC region”, in the Materials and methods section “Enrichment of eQTLs for regulatory elements” and in Figure 4—figure supplement 3.

Were they enriched for GWAS signals and have the eGenes in question been previous linked to disease?

We changed our approach to determine associations between eQTLs and complex traits to determine whether eQTL signals in the MHC region, rather than single variants, were associated with any of the 4,083 traits described in the UK BioBank. Using this approach, we used a colocalization approach(Giambartolomei et al., 2014) to test all the eQTL signals with each trait in the UK BioBank and observed 180 associations with PPA > 0.8 (Supplementary file 8, Figure 5). These results demonstrate the utility of our data to serve as a resource to narrow down the location of causal regulatory variants underlying known associations between genes in the MHC region and complex traits, as well as to identify biologically meaningful novel associations.

We have added a section in the Results (“Regulatory variants in the MHC region play important roles in complex traits”), in the Materials and methods (“Associations between eQTLs and GWAS”), Supplementary file 8 and Figure 5.